# Nonlinear mechanics of lamin filaments and the meshwork topology build an emergent nuclear lamina

K. Tanuj Sapra 1,2✉, Zhao Qin3,5, Anna Dubrovsky-Gaupp1, Ueli Aebi4, Daniel J. Müller 2, Markus J. Buehler3 & Ohad Medalia 1✉

The nuclear lamina—a meshwork of intermediate filaments termed lamins—is primarily responsible for the mechanical stability of the nucleus in multicellular organisms. However, structural-mechanical characterization of lamin filaments assembled in situ remains elusive. Here, we apply an integrative approach combining atomic force microscopy, cryo-electron tomography, network analysis, and molecular dynamics simulations to directly measure the mechanical response of single lamin filaments in three-dimensional meshwork. Endogenous lamin filaments portray non-Hookean behavior – they deform reversibly at a few hundred picoNewtons and stiffen at nanoNewton forces. The filaments are extensible, strong and tough similar to natural silk and superior to the synthetic polymer Kevlar®. Graph theory analysis shows that the lamin meshwork is not a random arrangement of filaments but exhibits small-world properties. Our results suggest that lamin filaments arrange to form an emergent meshwork whose topology dictates the mechanical properties of individual filaments. The quantitative insights imply a role of meshwork topology in laminopathies.

[1] Department of Biochemistry, University of Zurich, Winterthurerstrasse 190, 8057 Zurich, Switzerland. [2] Department of Biosystems Science and Bioengineering, ETH Zurich, Mattenstrasse 26, 4058 Basel, Switzerland. [3] Laboratory for Atomistic and Molecular Mechanics, Massachusetts Institute of Technology, Cambridge, MA 02139, USA. [4] Biozentrum, University of Basel, CH-4056 Basel, Switzerland. [5] Present address: Department of Civil and Environmental Engineering, Syracuse University, Syracuse, NY 13244, USA. ✉email: tanuj.sapra@bsse.ethz.ch; omedalia@bioc.uzh.ch

During the life cycle of a cell, the nucleus experiences large mechanical variations and sustains fluctuating deformations and stress[1–3]. The nuclear envelope (NE), comprising the outer (ONM) and inner nuclear membranes (INM) and the lamina underlining the INM[4], forms a rigid-elastic shell protecting the genetic material[5,6]. The nuclear lamina—a filamentous protein meshwork at the interface of chromatin and the nuclear membrane[4,7]—functions as a scaffold for binding transcription factors[8,9] and provides mechanical stability to the nucleus[10–13]. The principal components of the lamina in mammalian cells are mainly four lamins: lamins A and C, and lamins B1 and B2 (refs. [14,15]). Lamins are classified as type V intermediate filaments (IFs), and share a conserved tripartite domain structure with other IFs, viz., a central α-helical coiled-coil rod domain flanked by a non-helical N-terminal head and an unstructured C-terminal tail domain that hosts immunoglobulin (Ig) domains[16].

Mutations in lamins cause an important group of diseases termed laminopathies[17] that affect the load-bearing tissues such as striated muscles leading to mechanical failure[18,19]. It is therefore important to understand the underlying principles of lamin mechanics in health and disease[20,21]. Intact nuclei of Xenopus laevis (X. laevis) oocytes[22] and human fibroblasts were previously probed mechanically using the atomic force microscope (AFM)[23,24] to understand the role of lamina in nuclear mechanics. Micropipette aspiration has successfully provided deformation characteristics of nuclei and the NE[25,26]. Intact cells and whole organisms have also been employed to quantify the physical properties of the nuclear lamina using stretchable substrates[10,27] and microfluidic devices[28]. In an attempt to measure lamin mechanics directly, a sharpened tip of an AFM cantilever was pierced inside the nucleus through the plasma and nuclear membranes[24].

Studies performed on intact nuclei and entire organisms are informative but are influenced by the nuclear membranes, chromatin, and surrounding cells. Direct mechanical interrogation of native lamin filaments remains a pertinent goal towards understanding the mechanical properties of the lamina and the nucleus in health and disease[18,29–31].

In this work, using a combined mechanical, structural, and simulation approach, we characterize in situ assembled lamin filaments by applying point loads and measure their deformation and apparent failure in the native meshwork (Fig. 1). Our AFM-based force–extension (FE) measurements reveal that lamin filaments deform reversibly at low loads (<500 pN) followed by stiffening that culminated in failure at >2 nanoNewton (nN). The in vitro mechanical behavior of lamin filaments is recapitulated in silico by molecular dynamics (MD) simulations of single filaments in a meshwork model[32] derived from cryo-electron tomography (cryo-ET) of the nuclear lamina. A key finding is that the meshwork topology influences lamin filament mechanics. The study provides a general understanding of lamin filaments and nuclear lamina mechanics relevant to laminopathies.

## Results

### In vitro mechanics of lamin filaments assembled in situ.
Lamin dimers assemble into staggered head-to-tail polar structures[33,34] that interact laterally to form tetrameric filaments[35,36]. The filaments form a dense meshwork attached to the INM[21] at the nuclear lamina[4,7]. However, most lamins assemble into paracrystalline fibers in the test tube[4,36]. Therefore, native lamin filaments can only be studied using meshworks formed in situ.

First, we set out to visualize and probe lamin filaments in the nuclei of mouse embryonic fibroblasts (MEFs) and HeLa cells by AFM. Owing to the size of the mammalian nucleus (10–15 μm

diameter), its complexity, and tight connections of the chromatin with the lamina, opening a "window" into a mammalian nucleus is a daunting task. For this purpose, we developed a two-step procedure: in the first step, the cell membrane was de-roofed and the nucleus exposed and in the second step, the nuclear membrane was de-roofed opening windows into the nucleus and the lamin meshwork imaged (Supplementary Fig. 1a, b and "Methods"). Nuclei from both MEFs and HeLa cells that were chemically fixed (paraformaldehyde or methanol) during de-roofing showed a filament meshwork reminiscent of the lamin meshwork observed in cryo-ET of MEFs nuclei[4] (Supplementary Fig. 1c–g). In nuclei that were not chemically fixed during de-roofing, filamentous meshwork was not observed (Supplementary Fig. 1h, i) hindering visualizing lamin filaments and characterizing their mechanical properties.

Owing to its sheer size (~400 μm diameter) and a condensed chromatin structure that is not associated with the lamina[5], we utilized the X. laevis oocyte nucleus for AFM measurements of lamin filaments assembled in situ. Natively assembled meshwork of single B-type lamin (lamin LIII) filaments has been visualized by electron microscopy in spread NEs of the X. laevis oocytes[7,37,38]. In this study, we placed oocyte nuclei on a poly-L-lysine-coated glass surface to firmly attach them prior to manual opening with a sharp needle and washing away the chromatin. The procedure ensured that the cytoplasmic face of the nuclear membrane (i.e., the ONM) was attached to the poly-L-lysine surface and the lamin meshwork was exposed to the AFM cantilever tip (see "Methods"). The simple isolating and spreading of NEs from the oocytes in a physiological buffer without any detergent or fixative[39] enabled AFM and cryo-ET characterization of native lamin filaments (Fig. 1a). Removing chromatin by gentle washing could have removed some lamin-associated proteins but retained the farnesylated lamin LIII meshwork, allowing an unambiguous determination of the mechanical properties of lamin filaments in a meshwork[24,40].

The lamina was imaged by FE-based imaging (FE-imaging) with an active closed-loop feedback, recording FE curves at 128 × 128 pixels of the sample at a force of 0.5–0.75 nN. As observed previously by electron microscopy methods[4,7,38], lamin filaments were arranged in a meshwork exhibiting a rectangular pattern or a less organized architecture interspersed and interacting with the NPCs (Fig. 1b, c and Supplementary Fig. 2). After imaging a 1 μm × 1 μm area of the lamin meshwork, random points on lamin filaments were chosen (see "Methods") with an active closed-loop feedback. In the force-spectroscopy mode, the cantilever tip was pushed on those positions with a force of 5–8 nN perpendicular to the long axis of the lamin filaments (Fig. 1a, d). Since the diameter of the AFM tip is at least twice (~20 nm) the diameter of a lamin filament (~8 nm)[4], the tip applied force on the entire filament rather than at a specific position on the filament circumference. Thus, choosing the positions in the closed loop likely did not enable positioning the cantilever tip precisely at the center of the lamin filament. The ONM and INM of frog oocytes are ≈50 nm apart, while NPCs are ≈90 nm tall structures[41] with flexible lamin filaments situated on the nucleoplasmic face (Supplementary Fig. 3). This enabled pushing of lamin filaments up to 100 nm towards the glass surface (Fig. 1d, e).

The FE curves showed an initial slow rise in the force with a plateau up to a yield point (low-force regime) denoting a structural change where plastic or permanent deformation occured in the lamin filament. At larger strains, the filaments showed stiffening resulting in apparent failure at nanoNewton forces (high-force regime) (Fig. 1d). FE curves obtained upon pushing the cantilever with 5 nN force on areas of the NE without filaments did not show any plateau or peaks in the low- or the high-force regimes, respectively (Supplementary Fig. 4 and 5).

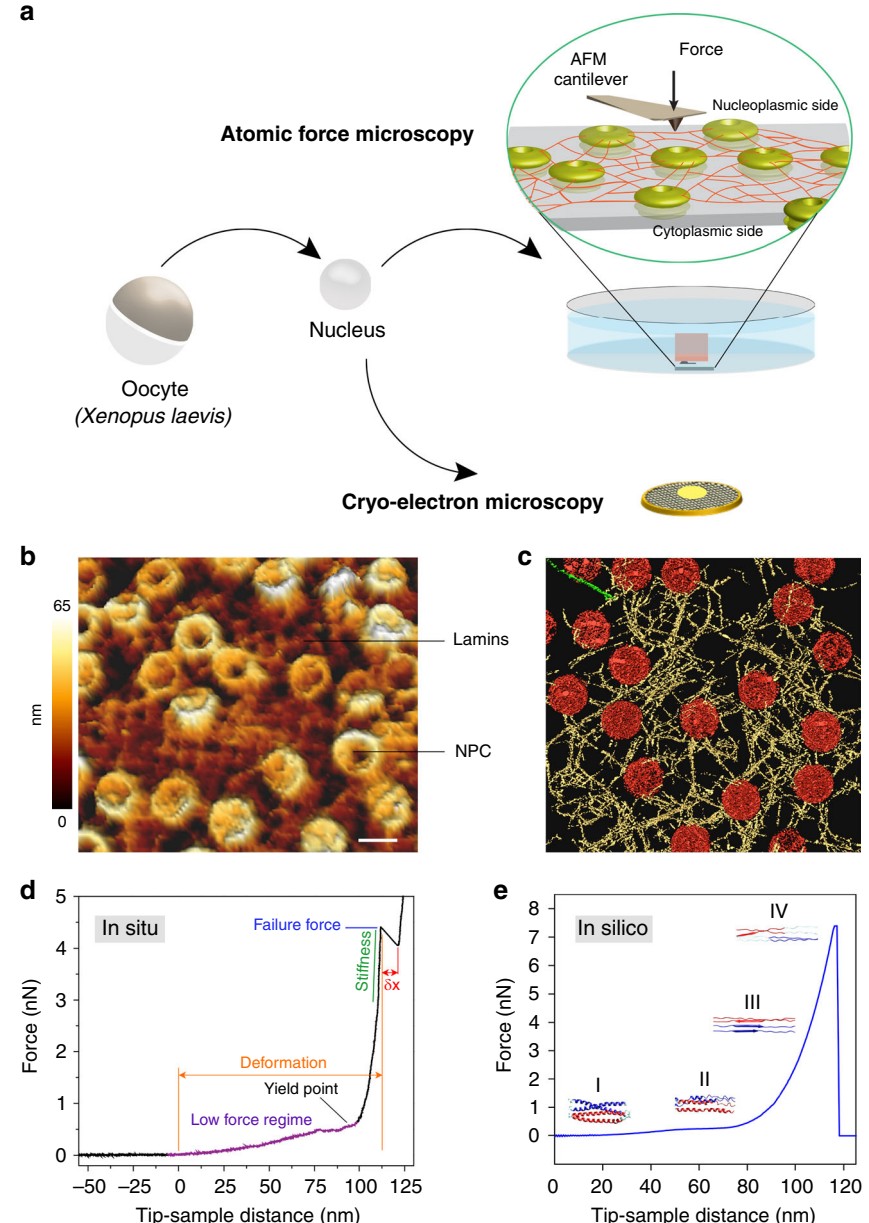

**Fig. 1 Revealing the structural mechanics of in situ-assembled lamin filaments. a** A schematic illustration of the experimental set-up. Isolated nuclei from *X. laevis* oocytes were attached onto a poly-ʟ-lysine-coated glass dish or carbon-coated electron microscopy grid. Next, the nuclei were manually opened by a sharp glass needle, and chromatin removed to expose the nucleoplasmic face. Zoom showing the cantilever tip pushing onto a lamin filament. For cryo-ET, an entire nucleus was collapsed on a grid by puncturing and removing the chromatin. **b** AFM imaging (nominal tip radius ≈10 nm) with a force of 0.5 nN. A representative image from the nucleoplasmic side showing areas of near-orthogonal arranged lamin filaments interspersed with NPCs. Randomly arranged filaments were also observed (Supplementary Fig. 2). Mechanical measurements on lamin filaments were performed on both meshwork types ($N = 51$, $N$ is the number of independent experiments). Color scale denotes the topograph height. Scale bar, 100 nm. **c** Surface-rendered view of a cryo-electron tomogram ($n = 13$ tomograms) acquired on a spread NE of the *X. laevis* oocyte. Nuclear lamins formed a 3D meshwork of filaments (yellow) connected to NPCs (red). Field of view, 700 nm × 700 nm. **d** A typical FE signal showing the nonlinear behavior of a lamin filament in the meshwork. **e** In silico, a single lamin filament in the meshwork when subjected to mechanical push showed a comparable FE profile; a low-force regime and a steep rise were identified indicating strain-induced stiffening leading to failure of the lamin filament. The different regions were assigned to the molecular changes in the lamin α-helical coiled coils. The yield point denotes the point of plastic or permanent deformation, i.e., irreversible structural change. The similarity between the FE curves obtained in vitro and in silico suggests that single lamin filaments were probed.

The nonlinear strain-stiffening behavior observed here for lamin LIII resembles previously described properties of other filamentous proteins[42–44] including IFs[45,46], and rather reflects a general mechanism of material deformation and failure[47]. Strain hardening was also reported for reconstituted lamin B networks and is dramatically higher than observed in networks of keratin, vimentin, or F-actin[48].

To elucidate the structural intermediates during the mechanical pushing of a lamin filament, we performed molecular dynamics (MD) simulations on single filaments in a meshwork model derived from cryo-electron tomograms of the nuclear laminae of *X. laevis* oocytes and MEFs (see "Methods"). Mechanical behavior of single lamin filaments was simulated by applying an out-of-plane pushing force on single points in the

meshwork model. Similarities between the AFM and the in silico FE curves suggest that we probed single lamin filaments in vitro (Fig. 1d, e). The simulations showed that the plateau in the low-force regime in the FE curves is due to partial unfolding of the α-helical coiled-coil domains. However, sliding between lamin dimers cannot be excluded although direct interactions of the farnesylated lamin tails with the INM makes this less likely. It should be noted that in MD simulations performed at a few orders of magnitude higher speeds than experiments, we did not observe clear sliding. Sliding may occur more frequently at lower speeds (as in experiments) as at higher speeds (»1 μm s$^{-1}$) sliding requires much larger forces than unfolding coiled-coil α-helices making unfolding the dominant event[49]. The inter-connectedness of the filaments at junctions may also resist sliding and promote strain-induced stiffening. The stiffening at high force represents the transition of α-helical coiled coils to β-sheets followed by failure. The force tolerated by a filament before failure in silico was comparable to that measured on lamin filaments assembled in situ. Simulations showed that the FE curve profiles did not depend on the size of the probe used to push the lamin filaments, and the transition from α-helix to β-sheet occurred through similar structural intermediates (Supplementary Fig. 6a–c). It is suggested that the mechanical reaction of lamin filaments is a robust characteristic and may be key to its function under different mechanical loads; for example, during cell migration or during NE breakdown[13,48] when the nucleus experiences loads.

Single IFs were shown to withstand nanoNewton forces when pushed perpendicular to the long axis[50], stretched along the long axis[51,52], or laterally dragged on a surface[53]. Interestingly, these studies showed that laterally pushing on cytoplasmic IF proteins in vitro required forces of 3–5 nN to break and low- and high-force regimes resembling our observation that lamin filaments can withstand forces of up to a 3–5 nN. Previous MD simulations of stretching lamins in an orthogonal meshwork[32] (Supplementary Fig. 7) also showed nonlinear stress–strain profiles and failure forces similar to those observed by mechanical pushing of in situ-assembled lamin filaments (Fig. 1d).

**Detailed mechanical behavior of lamin filaments.** The FE curves recorded in vitro showed a plateau (15–35%) in the low-force regime (Fig. 2a and Supplementary Table 1) at a force $F_{low} \approx 0.3$ nN (Fig. 2b). The plateau presumably denotes unfolding and sliding of an α-helical coiled coil preceding its conversion into stiffer β-sheets[49] (Fig. 1e and Supplementary Fig. 6). The "soft" coiled coils ($\kappa_{low} \approx 0.008$ nN nm$^{-1}$, (Fig. 2c)) experienced entropic stretching up to a deformation $d_{low} \approx 54$ nm (Fig. 2d). An engineering strain of ≈126% (Supplementary Note 1) before stiffening was observed similar to other semi-flexible filaments of comparable persistence and contour lengths[4,54]. Unfolding of α-helical coiled coils before transition into β-sheets is a well-known phenomenon[55], and has been suggested for vimentin[56], fibrin, and fibrinogen during blood clotting[44,57].

Beyond the yield point and prior to failure, an intermediate peak (4–16%) of step unit ≈1.9 nm was detected between 1.0 and 1.5 nN (Fig. 2e–h and Supplementary Table 1). A step unit of ≈1.9 nm corresponds to the diameter of individual lamin dimers (≈1.7 nm)[58] (Supplementary Fig. 8). We deduced that the intermediate peak denotes the rupture of dimer–dimer interactions during the transformation from folded α-helices to β-sheets.

In the high-force regime beyond the yield point (Fig. 1d), lamin filaments showed plastic deformation. Plastic deformation was not observed upon indenting nuclei most likely because of limited indentation depths and interference from the NE hindering direct contact with the lamin meshwork[25]. The FE curves showed an abrupt drop in force with step units $\partial x_{high} \approx 4$ or 8 nm (Fig. 2i)

between $F_{high} \approx 1.5$–5.0 nN (Fig. 2j). Due to contact mechanics of the cantilever tip and the lamin filament, there may be a lateral force acting on the filament which may lead to tip slippage from the filament. Therefore, the forces measured are probably an underestimation of the forces that lamin filaments are capable of withstanding.

Corroborating the origin of the step units, cryo-ET analysis of the oocyte NE confirmed that lamin tetramers of diameter 3.8 nm associate laterally to form filaments of diameter 7 nm (bundling of lamin filaments) (Supplementary Fig. 8a, arrowheads, c)[4,7]. MD simulations suggest that in vitro filament stiffening, $\kappa_{high} > 0.3$ nN nm$^{-1}$ at high forces (Fig. 2k), is caused by α-helix coiled-coil unfolding and transitioning to β-sheets. The α-helix to β-sheet transition in lamins is also based on previous experimental studies on hagfish slime thread[46], hard and soft α-keratins[45,59], and vimentin[52].

With a capacity to withstand an average deformation $d_{high} \approx 91$ nm (up to 200 nm) (Fig. 2l), lamin filaments showed a remarkable engineering strain of $\varepsilon \approx 250\%$ (Supplementary Note 1) similar to the average strain of ~250% observed for single IFs[53,60]. The strain of lamin filaments measured here is higher than that of collagen (12%), α-keratin (45%), elastin (150%)[61], but comparable to desmin (240%)[50], vimentin (205%)[52], and hagfish slime (220%)[46].

Increasing the loading rate over two orders of magnitude did not vary the step unit, force, deformation, or the stiffness of lamin filaments (Supplementary Fig. 9 and 10). Hysteresis between the approach and retract FE curves suggests that the lamin filament has viscous characteristics (Fig. 3a). However, the filaments stiffened to a similar extent at increasing loading rates indicating the absence of viscous structural elements (Supplementary Fig. 11), and that the energy (denoted by the hysteresis) is stored in the β-sheets[56]. Interestingly, a direct correlation between the failure force and the stiffness implies that as a filament stiffens, its capacity to bear load increases (Supplementary Fig. 12).

Following the first peak, ≈60% of the FE curves also showed a second peak with a step unit of ≈4 nm (Supplementary Fig. 10 and 13), while additional peaks were seldom seen as well (Supplementary Fig. 14). Akin to the first peak, the second peak presumably indicates the failure of an additional lamin tetramer that was also detected by cryo-ET (Supplementary Fig. 8c) although the second and subsequent peaks may be a result of cantilever slippage. However, the peaks after the first peak also occurred at nanoNewton forces and with decreasing frequency (Supplementary Fig. 14 and Supplementary Table 3) suggesting that the peaks represent lamin failure as the tip cannot slip multiple times from the same filament. Alternatively, these peaks may detect interactions between lamins (bundling) and other binding proteins. Control experiments showed that the forces measured were specific to lamins since pushing directly on nuclear membranes produced FE curves without any prominent peaks (Supplementary Fig. 4). Moreover, characteristic force response of lamin LIII filaments was still obtained after nuclease treatment, and the measured parameter values were comparable to those obtained without nuclease treatment (Supplementary Fig. 15). Since nanoNewton forces are not typical of protein-protein interactions, and filaments were pushed at hundreds of positions, we reckon that the measured parameters are predominantly of the lamin LIII filaments.

**Lamins are tough and act as shock absorbers.** The lamin meshwork is suggested to function as a shock absorber protecting the nuclear contents from external mechanical forces[5]. A question we asked was: is shock absorption an emergent property of the meshwork or is a single filament capable of absorbing shocks too?

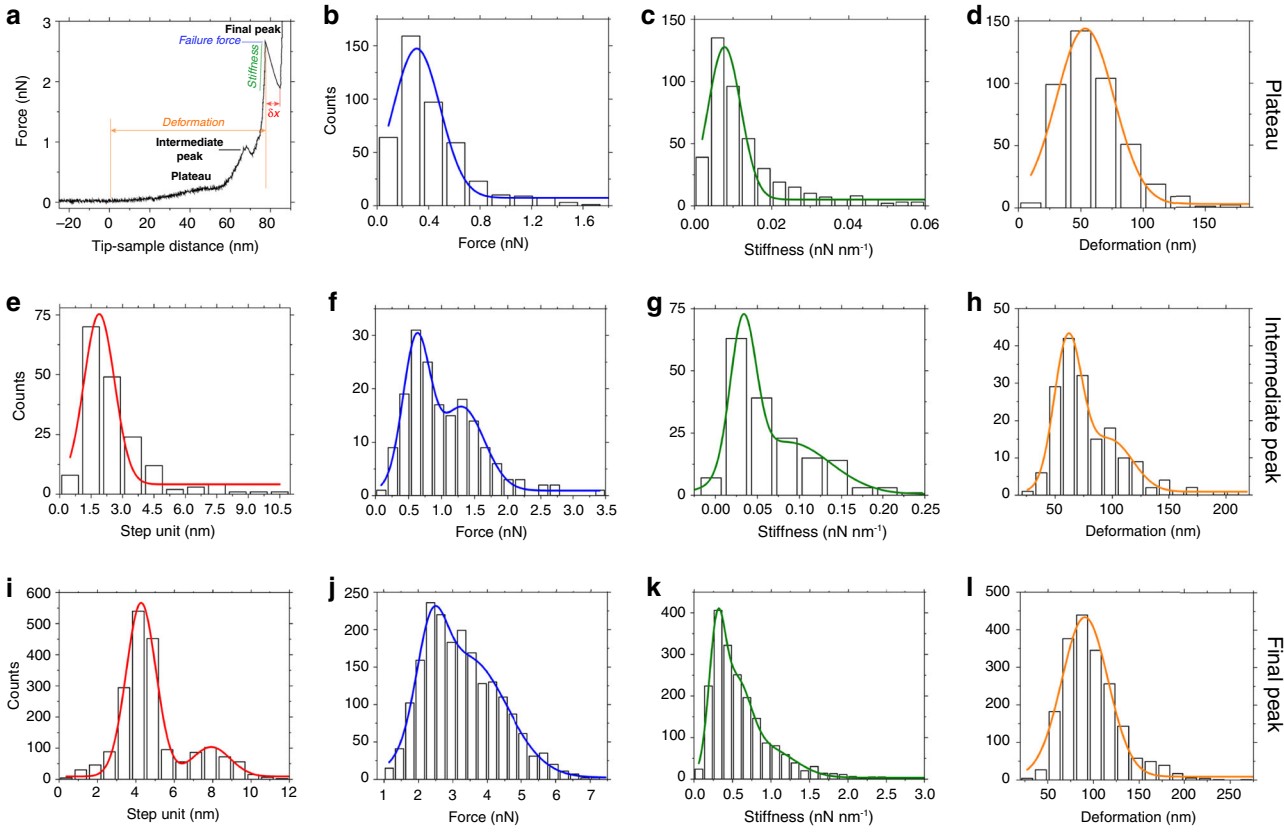

**Fig. 2 Mechanical characteristics of lamin filaments. a** An FE curve showing a characteristic plateau and an intermediate peak preceding the final peak at high force (≈3 nN). **b** The plateau occurred at a force of 0.30 ± 0.20 nN (average ± standard deviation calculated from FWHM). **c** The region showed a stiffness of 0.0078 ± 0.0043 nN nm$^{-1}$ ($n = 432$, $n$ is the number of single events), and **d** a deformation of 54 ± 23 nm ($n = 432$). **e** The step unit of the intermediate peak was 1.9 ± 0.7 nm at a force of 0.60 ± 0.20 nN ($n = 175$) (**f**). **g** A stiffness of 0.033 ± 0.015 nN nm$^{-1}$ ($n = 172$) was fivefold more than that of the structural intermediate at the plateau. **h** A deformation of 61 ± 12 nm ($n = 173$) was marginally larger than that at the plateau. The shoulders in the histograms, **f** force (1.3 ± 0.30 nN), **g** stiffness (0.086 ± 0.050 nN nm$^{-1}$), and **h** deformation (98 ± 20 nm) indicate a continuum of the stiffening process. **i** Step unit histograms (4.3 ± 0.80 nm; 7.9 ± 1.0 nm; $n = 1946$); **j** failure force (2.4 ± 0.40 nN; 3.5 ± 1.1 nN; $n = 1946$); **k** stiffness (0.30 ± 0.11 nN nm$^{-1}$; 0.54 ± 0.19 nN nm$^{-1}$; 0.95 ± 0.34 nN nm$^{-1}$; $n = 1946$); and **l** deformation (91 ± 26 nm; $n = 1946$) of the final peak data pooled from all speeds (Supplementary Fig. 10).

We directly measured the energy absorbing capacity of lamin filaments assembled in situ. For this, we applied a repetitive force protocol to measure the energy dissipated in the low-force regime (plateau) before stiffening. The first step of the protocol—pushing up to 0.5 nN—showed a plateau; however, the relaxation curve did not show a plateau (described previously for perfect spring proteins like myosin[62]). The second pushing step (up to 8 nN) showed the plateau again resulting in a hysteresis between the pushing and the relaxation curves (Fig. 3a). The recurrence of the plateau suggests that either the refolding or the spring-like sliding of the α-helical coiled coil[63] is reversible and occurs within ≈500 ms (see "Methods"). Interestingly, the plateau force did not depend on the loading rate (Supplementary Fig. 16) and did not recur after filaments had experienced nanoNewton forces up to the apparent failure.

The energy attributed to the hysteresis in the low-force regime was determined to be ≈10$^{-17}$ J (Fig. 3a, c) increasing to ≈10$^{-16}$ J (10$^5$ $k_B$T) up to the apparent failure (Fig. 3b, d). The energies did not change with loading rates providing evidence that lamin filaments in healthy state are ductile and not brittle[64]. A back-of-the-envelope calculation suggests that the total energy absorbed by the filaments during pushing is equivalent to that required for breaking ≈170 C–C bonds (1 C–C ≈ 5.8 × 10$^{-19}$ J). However, the

measured force ($F_{high}$) of 2–5 nN is similar to the strength of a single C–C bond[65]. Increasing the pushing speed of the cantilever 20-fold, and hence the kinetic energy imparted by the cantilever 400-fold, did not change the energy dissipated by lamin filaments. We therefore suggest, that the energy provided by the cantilever is not utilized for breaking covalent bonds but is expended in disrupting non-covalent interactions (sum of charges, van der Waals forces, and hydrogen bond interactions) involved in the unfolding of protein structures, including the force-induced α-helix to β-sheet transformation and the stick-slipping process of the β-sheets under tension[49].

Based on the total energy, the tensile toughness, $T$, of a lamin filament ($r ≈ 2$ nm) is estimated to be ≈147 MJ m$^{-3}$ (volume, $V ≈ 679$ nm$^3$) or ≈10$^5$ J kg$^{-1}$. Remarkably, the toughness of a lamin filament is superior to that of high-tensile steel (6 MJ m$^{-3}$), carbon fiber (25 MJ m$^{-3}$), and Kevlar 49 fiber (50 MJ m$^{-3}$), much higher than the toughness of natural materials such as elastin (2 MJ m$^{-3}$), resilin (4 MJ m$^{-3}$), tendon collagen (7.5 MJ m$^{-3}$), and at par with that of wool (60 MJ m$^{-3}$), *Bombyx mori* cocoon silk (70 MJ m$^{-3}$), nylon (80 MJ m$^{-3}$), *A. diadematus* dragline (160 MJ m$^{-3}$), and viscid (150 MJ m$^{-3}$) silks[66]. The strong, tough and extensible nature of lamin filaments offer promising possibilities for engineering lamin-based materials.

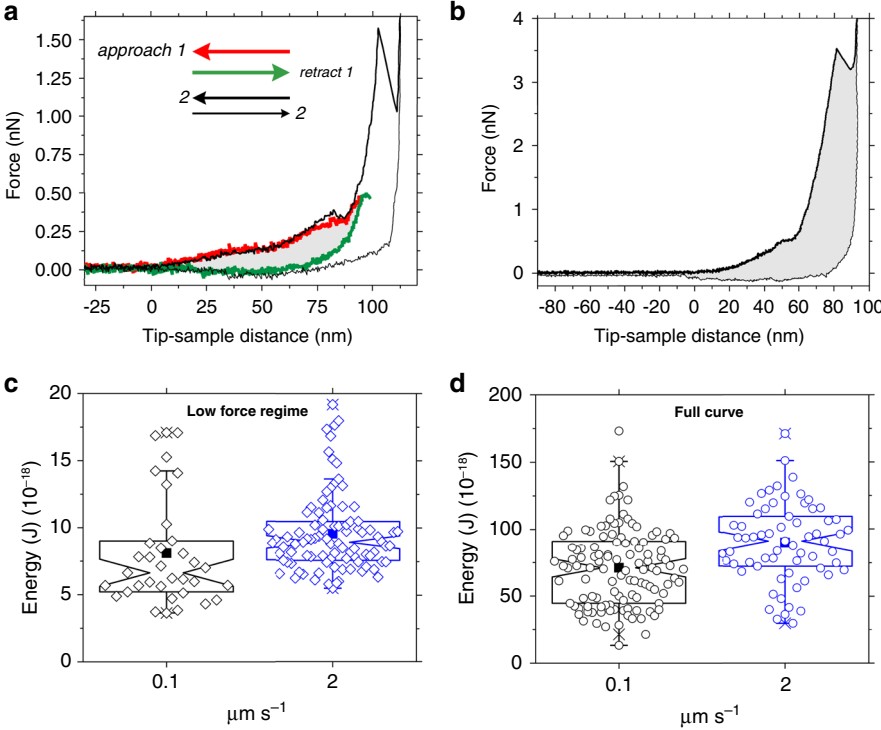

**Fig. 3 Lamin filaments absorb energy under continuous applied force. a** To estimate the energy absorbed during the initial stretching of a lamin filament (low-force regime), a repetitive force protocol was used. A force of 0.5 nN was applied (approach, red curve) and released (retract, green curve); the area between the red and green curves (gray) gives the energy absorbed by the lamin structure. The same filament was subjected to a higher force until failure was detected (bold black curve). The plateau in the low-force regime was reversible (compare black and red curves) in majority (89%) of the FE curves ($n = 133$). The reversibility shows that the cantilever tip could be stably and reproducibly placed on a lamin filament and that a single filament was interrogated. **b** The area (shaded gray) between the approach curve (bold black) and the retract curve (thin black) denotes the energy dissipated during the entire process of failure. **c, d** The energy absorbed in the low-force regime was $\approx 10^{-17}$ J ($n = 40$ at 0.1 µm s$^{-1}$, $n = 93$ at 2 µm s$^{-1}$, $n$ is the number of single experimental events as shown in **a, b**), an order of magnitude lower than that absorbed during the irreversible failure $\approx 10^{-16}$ J ($n = 115$ at 0.1 µm s$^{-1}$, $n = 65$ at 2 µm s$^{-1}$). Each notch-box includes 25–75% of the data, and each diamond and circle denotes a single event from a filament. The solid squares denote the mean and the whiskers signify 1–99% of the data.

**Lamin filaments can withstand constant forces upto nano-Newtons.** Is the mechanical behavior of lamin filaments peculiar to force loading at a constant velocity or could it be recaptured at constant loads too? To answer this, we subjected lamin filaments in the meshwork to constant loads ($F_{load}$) of 0.75–3.0 nN. As in constant velocity experiments, discrete steps of 1.3 nm ($F_{load} \leq$ 1 nN), 4 nm, or 8 nm ($F_{load} \geq 2$ nN) were detected (Fig. 4).

The lifetime, $\tau_{break}$, of a few hundred milliseconds measured for the α-helical coiled coils at $F_{load} \leq 1$ nN makes it the first buffer against mechanical shocks at lower forces. The coiled-coil structure absorbs the kinetic energy and prevents force propagation to further regions of the meshwork. As the force is increased to 3 nN, the α-helix to β-sheet transition increases the load-bearing capacity of the filament because of stiffening, and the failure requires tens of milliseconds (Supplementary Fig. 17). Local stiffening and failure at nanoNewton forces may serve as an efficient mechanism preventing breakage of other filaments and a catastrophic meshwork failure[32].

**Meshwork topology influences lamin mechanics.** Visually, the lamin meshwork appears as a random arrangement of filaments. Because a cryo-ET tomogram provides a 3D view of the overlapping areas of a meshwork, it is difficult to discern if the lamin meshwork is near-orthogonal or random (observed in 2D AFM and SEM images). Hence, we performed network analysis to bring out the hidden rules in the meshwork. The network analysis

also provided a framework to: (i) compare the design features of the lamin meshwork in different species, and (ii) decipher the influence of meshwork topology on lamin filament mechanics. In a first step, we employed cryo-ET for visualizing the nuclear laminae of *X. laevis* oocytes (Fig. 5a) and MEFs (Fig. 5e, cryo-ET data from ref. [24]). Next, we applied graph theory[67] to compare and quantitate the meshwork topologies. To this end, the meshworks were converted to undirected 3D graphs; lamin filaments formed the links between the vertices (nodes) representing physical connections between the filaments (Fig. 5b, f). Interestingly, the lamin meshworks of both, *X. laevis* oocyte and mammalian nuclei, exhibit similar topological features. The average degree of connectivity, <*k*>, of both the meshworks is 3.3. The degree (*k*) distribution of both the meshworks follows a power-law [$P(k) \sim k^{-\lambda}$] with a scaling factor $\lambda > 5$. In other words, the meshworks consist of many nodes with connectivity 3 or 4 and a minor population of hubs of high degrees (5–17) (Fig. 5c, g). It is noteworthy that despite the difference in lamin types (B-type lamin LIII in *X. laevis* oocyte, and A-type and B-type MEFs lamins), the lamin meshworks of both species share common topological features. Furthermore, in both meshworks, the average path-length scales with the meshwork size, i.e., the distance between the nodes scales with the number of nodes (Fig. 5d, h). This "small-world" property of the lamin meshwork, defined simply as the nodes connected through the shortest distances, is similar to that of a power grid network[68] and points to the importance of hubs to the integrity of the meshwork. Although

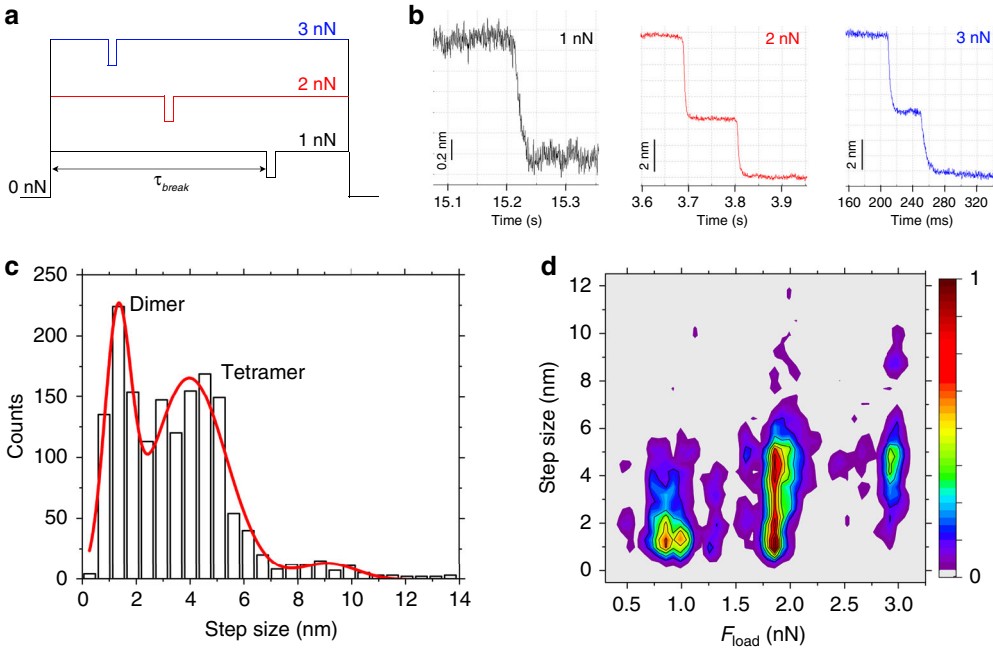

**Fig. 4 Discrete failure steps of lamin filaments at constant loads (force clamp). a** Lamin filaments were subjected to constant loads ($F_{load}$) ranging from 0.75 to 3 nN. The valleys denote a drop in force at break events. $\tau_{break}$ denotes the lifetime of a filament at a certain force (Supplementary Fig. 17). **b** Typical signals denoting the molecular alterations in lamin filaments under constant loads ($F_{load}$) of 1, 2, and 3 nN. **c** Failure of lamin filaments at constant $F_{load}$ occurred in discrete steps of 1.3 ± 0.5 nm, 4.0 ± 1.4 nm, or rarely 9.0 ± 1.1 nm ($n = 1569$, $n$ is the number of single events as shown in **b**). **d** A density map showing that the main population consists of steps ≈1 nm at low loads (≤1 nN) increasing to ≈4 nm at high loads (≥3 nN) with a transition observed at 2 nN. The 1.3 nm step is attributed to the mechanical rupture of the α-helix coiled coil (diameter ≈1.2–1.7 nm)[4,58]. The 4 nm steps denote the failure of stiffened tetramers at a high force. Color scale denotes normalized densities of the populations.

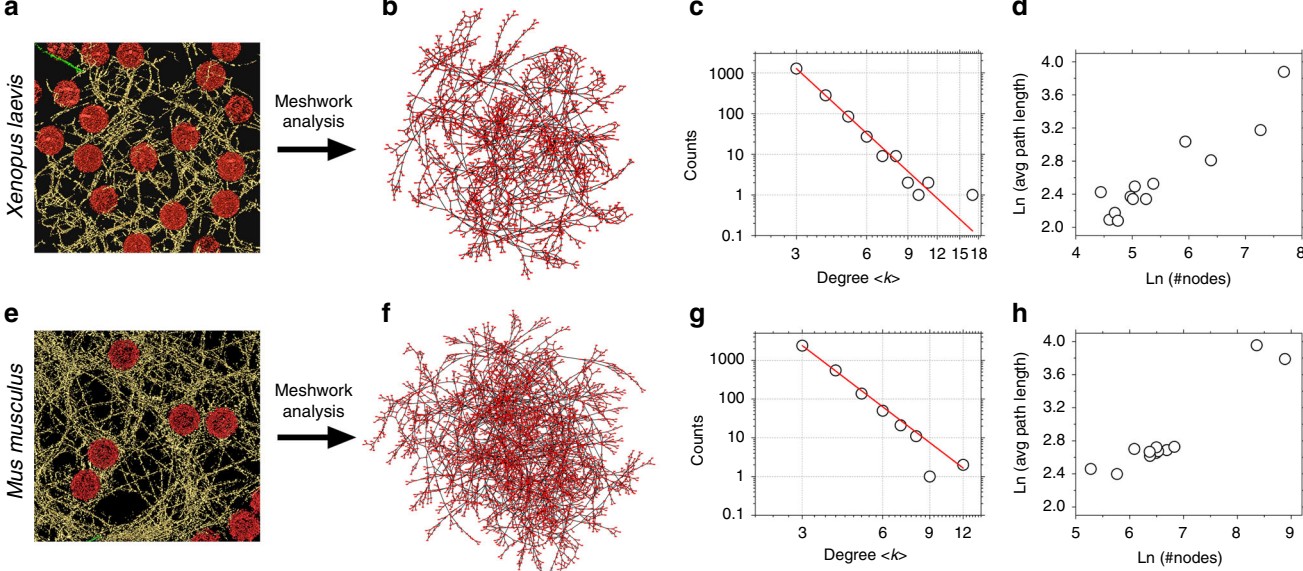

**Fig. 5 Nuclear lamin meshworks of *X. leavis* oocyte and mouse embryonic fibroblast show similar topology.** The 3D lamin meshworks as viewed by cryo-ET of **a** *X. laevis* oocyte NE ($n = 13$ tomograms) (field of view, 700 nm × 700 nm) (Fig. 1b) and **e** MEF NE ($n = 12$ tomograms) (adapted from ref. [4]; field of view, 1000 nm × 1000 nm) were analyzed to create undirected graphs (**b**, **f**). Nuclear lamins formed a 3D meshwork of filaments (yellow) connected to NPCs (red). The red dots in the graphs denote the nodes or the vertices (coordinates of filament interactions or crossovers) defined as the points where adjacent lamin filaments (gray connecting lines) appeared closer than 1.3 nm. **c**, **g** Degree distribution of both the lamin meshworks revealed a power-law behavior with an index $\lambda \approx 5.6$, and **d**, **h** exhibited "small-world" characteristics where the average path-lengths scale with the meshwork size.

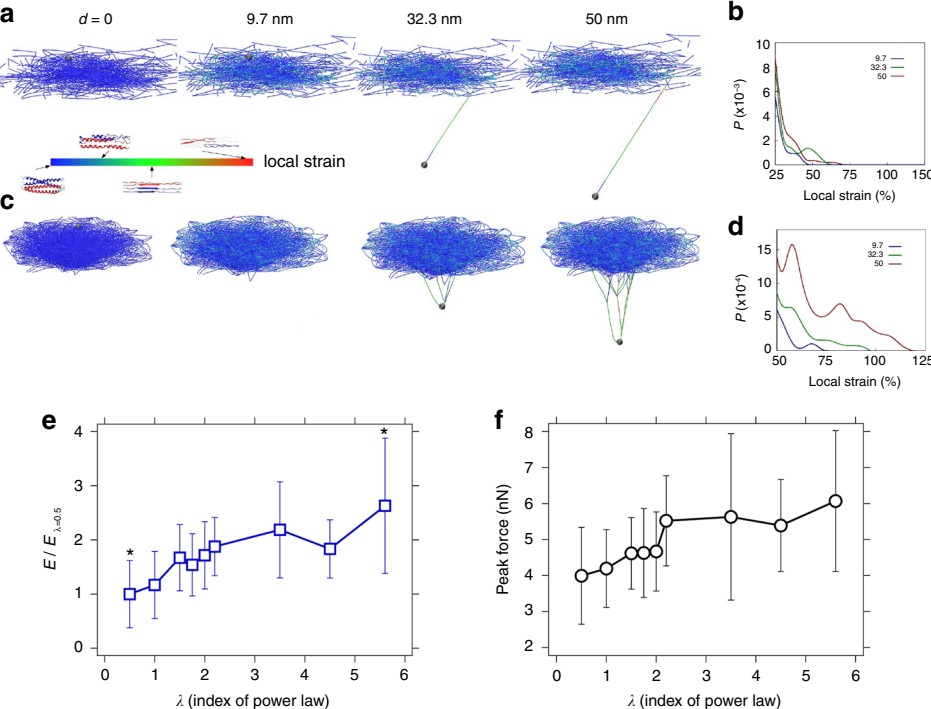

**Fig. 6 Meshwork topology ($\lambda$) influences toughness and strength of lamin filaments.** Snapshots from simulations of mechanical pushing of single lamin filaments in meshworks of two different node connectivities: **a** $\lambda = 0.5$ (Supplementary Movie 1) and **c** $\lambda = 5.6$ (Supplementary Movie 2). The meshwork of $\lambda = 0.5$ was composed of many single filaments connected to a few heavy nodes, while the meshwork of $\lambda = 5.6$ was composed of well-connected filaments with a balance of light nodes, intermediate nodes and a few heavy nodes. The color scale denotes the local strain at different deformations, $d$, of the filament. **b, d** Probability distributions of the local strain within each filament segment measured at increasing deformations (corresponding to snapshots in panels **a, c**) for the two meshworks. At $\lambda = 0.5$, the filament being pushed ruptured from one end at an early stage ($d < 32.3$ nm), the other end was stretched and unraveled until the heavy node and the strain in the filament increased to cause rupture. At $\lambda = 5.6$, increasing deformation caused a large strain re-distribution. The meshwork with $\lambda = 5.6$ was associated with a much larger cohesive zone that enabled to dissipate more deformation energy before rupture (see "Methods"). Lamin filaments in meshworks with higher $\lambda$ values were significantly **e** tougher ($E$) and **f** stronger as compared to filaments in meshworks with smaller $\lambda$. Comparing the values from $\lambda = 0.5$ and $\lambda = 5.6$ with a $t$-test yielded a $p$ value of 0.05 denoted by *. Each mean value and standard deviation in **e** and **f** are obtained from FE curves of 100 independent simulations on networks of the same $\lambda$ value, as summarized in Supplementary Fig. 18.

we cannot elucidate the structural identity of the lamin connections (nodes), cross-linking mass spectroscopy studies suggest that electrostatic interactions between the unstructured head and tail domains of adjacent lamin dimers may drive meshwork assembly[63].

Mutations in lamins and NE proteins[69] are suggested to influence the mechanical properties of the nucleus[70,71]. We conjectured that the meshwork topology may influence the mechanical properties of the lamina. To test this hypothesis, we performed MD simulations for hundreds of times by pushing single filaments in meshwork models of different topologies (Supplementary Fig. 18). Meshwork topologies were created by changing the $\lambda$ values between 0.5 and 5.6; filaments in a meshwork of $\lambda = 0.5$ were the least connected and $\lambda = 5.6$ were the most connected (see "Methods"). For all meshwork topologies, the simulated profiles of the FE curves resembled those obtained in vitro (Fig. 1c).

Interestingly, the strain propagation along the filaments and the filament strength increased with the meshwork connectivity. At $\lambda = 0.5$, increasing filament deformation caused a localized change in the meshwork strain and filament failure occurred at smaller strains (Fig. 6a, b and Supplementary Movie 1). At $\lambda = 5.6$, increasing filament deformation led to a larger strain propagation in the meshwork peeling off the filaments from the

underlying meshwork followed by local filament failure at higher strains (Fig. 6c, d and Supplementary Movie 2). Furthermore, in a highly connected meshwork, filaments underwent larger deformation and were able to sustain higher forces as compared to a less connected meshwork (Supplementary Fig. 19). For each $\lambda$ value, we regenerated the network and repeated the pushing simulation on a randomly selected filament near the middle of the network (to reduce free edge effects of the network model) for 100 times (Supplementary Fig. 18). We measured the strength (failure force, defined as the maximum force value of an FE curve) and toughness (energy dissipated or stored, as the total area below an FE curve) each time and summarized their average values and standard deviations as the plot points and error bars, respectively, as shown in Fig. 6e, f.

Biological phenomena exhibiting emergent characteristics have been reported at the cellular[72,73] and molecular[74,75] scales including for cytoskeletal filaments[76]. Our results suggest that the lamin meshwork is an emergent structure—the meshwork is more than the sum of its parts. The force required to damage an entire lamin meshwork by an outward pressure was determined to be $\approx 300$ nN[77], and the work done on the meshwork was estimated to be $\approx 10^8$ $k_B T$. The high values on the meshwork compared to a single filament ($\approx 10^5$ $k_B T$) indicate the emergent nature of the meshwork may be a key design feature.

**Mechanics of mammalian nuclei**. Our MD simulations and previous work[78] suggest that the lamin meshwork topology determines filament and nuclear mechanics. To test this, we measured the resistance or counter-force of isolated MEFs nuclei by confining them between two parallel surfaces—a flat-wedged AFM cantilever and a glass surface[79] (Supplementary Fig. 20). We observed that, (i) the counter-force generated by a nucleus on the cantilever increased as the confining space decreased, and (ii) nuclei with either lamin A alone (lamin B knock-out) or lamin B alone (lamin A knock-out) showed higher counter-force than the wild-type nucleus (Supplementary Fig. 21). Our results indicate that nuclear mechanics depend on the relative concentrations of the major lamins which may also influence the meshwork topology[80] (Supplementary Fig. 1f, g).

## Discussion

Mechanical measurements of isolated nuclei[6,25,69], intact cells[23], or entire organisms[27] by micropiperte aspiration, AFM, and stretchable substrates provide insight into nuclear stiffness and morphology[81]. Such studies have assessed the changes to differing levels of lamins and NE proteins, and the underlying alterations of the physical properties[21]. Here, we characterized the mechanics of in situ assembled lamin filaments and meshworks providing a close-to-native view of the physical properties of nuclear lamins. However, the measurements were conducted using isolated NEs from oocytes, and the filaments in vivo could exhibit differences in assembly and physical properties from those observed here because of local molecular crowding or osmotic effects[82]. Nuclei treated with detergents or different buffers for cryo-ET[4] and super-resolution microscopy (3D-SIM)[80] show similar lamin meshworks; however, opening the nucleus and washing away the chromatin and associated proteins may have an effect on the lamin meshwork organization and the mechanical properties. Even if the meshwork is altered in our system, our work focuses on the mechanical properties of individual lamin filaments and shows that the meshwork topology dictates the mechanical properties of the lamina. This is a purely physical phenomenon and suggests that different lamin meshworks exhibit varying mechanical properties in agreement with previous studies with overexpression of lamins in different tissues[83] and also in disease states[71]. Our study bridges the gap between in vitro and in vivo by mechanically characterizing the filaments in a lamin meshwork assembled in situ but without the influence of chromatin, nucleoplasm components and the nuclear membrane[24,40].

Combining mechanical and structural tools for interrogating individual lamin filaments offers a glimpse into the molecular mechanisms responsible for the mechanical properties of the lamin meshwork. The direct mechanical measurements of lamin filaments and meshwork simulations provide a mechanistic glimpse into their role in protecting the nuclear contents in response to external forces (Fig. 7). The nonlinear behavior of lamin filaments under applied load and their connectivity in the meshwork confers exceptional strength and strain to the filaments. The filaments resist mechanical load by their stretching capacity rather than immediate breaking.

Mechanisms protecting the lamin meshwork against mechanical force are integrated at each level in the hierarchical construction of the meshwork—starting from the basic building block α-helix coiled coil up to the higher order meshwork (Fig. 7b). The reversible unfolding or even sliding of the α-helical coiled coil (rod domain) at low forces is the first protective step buffering mechanical shock given to a nucleus ensuring the structural integrity of the lamina and the nuclear contents (Fig. 7b, middle). At high forces, an irreversible strain-induced stiffening increases the filament strength presumably by α-helix to

β-sheet transition further fortifying the meshwork against failure (Fig. 7b, bottom). We emphasize that our model is based on in vitro measurements of in situ assembled lamin meshwork and would require further investigation to prove its validity in vivo.

Interestingly, the strength (failure force), toughness (energy dissipated or stored) and the resilience of the filaments increased as additional hubs were introduced in a meshwork (increasing λ from 0.5 to 5.6) (Fig. 6e, f). Our results correlate the lamin meshwork topology and mechanical properties of single filaments, and may explain how re-modeling of the lamin meshwork may modulate nuclear mechanical properties, mechanotransduction and gene regulation[84].

Here, we used the interactome of a subset of the nuclear lamin meshwork to model and simulate the meshwork mechanics for a direct comparison with experiments. The combination of AFM and cryo-ET together with network analysis opens possibilities to understand the structural basis of nuclear mechanics in health and disease[78]. The application of network theory can be applied to quantitatively correlate the organization of the nuclear lamina to its mechanical role in diseases and malfunction by debilitating mutations[71] in cells and tissues[83]. The integrative approach is a promising step towards combining structural mechanics and visual proteomics of entire cells and cellular organelles for obtaining insights into the function of macromolecular assemblies in health and disease[85]. The results also have implications for material applications of lamins similar to silk and Kevlar®, and rationale design of protein-based meshworks with advanced mechanical functions[86]. For example, protein engineering of lamins combined with 3D printing technologies could be an area to explore in the near future.

## Methods

**Xenopus laevis nuclear lamina preparation for AFM measurements**. X. laevis oocytes at stage VI were allowed to swell in a low-salt buffer (LSB) (10 mM HEPES, 1 mM KCl, 1 mM MgCl₂, pH 7.4) for 20–25 min. A prick with a sharp needle punctured the oocyte and enabled the nucleus to slowly squeeze out. The intact nuclei were immediately transferred to Modified Barth's Buffer (MBB) (7.5 mM HEPES, 88 mM NaCl, 1 mM KCl, 0.4 mM CaCl₂, 0.8 mM MgSO₄, 2.5 mM NaHCO₃, 2 mM Ca(NO₃)₂, TRIS to pH 7.5) and washed gently by a stream of the surrounding buffer repeatedly. The nuclei were then transferred to another Petri dish (World Precision Instruments) coated with poly-L-lysine (1 mg mL⁻¹) which enabled the nuclei to stick firmly onto the glass surface of the dish. With a glass microneedle the nucleus was slightly pushed onto the surface while rolling the needle to break open the nuclear membrane such that the nucleoplasmic side was facing upward, i.e., INM. The nuclear contents including chromatin were gently removed and the stuck nuclear membrane washed with an ample volume of MBB (10–15 mL). If the ONM, i.e., the cytoplasmic side, were facing upward, we would not expect to see lamin filaments but only NPCs when imaged by AFM[87]. For the experiment with Benzonase® nuclease (Merck), the open nuclear membrane was incubated with 2500 U mL⁻¹ of the nuclease for 1–2 h at room temperature.

**HeLa and MEFs nuclear lamina preparation**. Nuclear lamina of HeLa Kyoto and MEF cells were prepared for imaging with AFM by de-roofing the nuclei. Cells were seeded on autoclaved coverslips (#1 or 1.5, Carl Roth) and allowed to grow at 37 °C (5% v/v CO₂) until a confluency of 75–90 % was obtained. The cells were prepared by washing the coverslips first with Ringer's solution (+2 mM CaCl₂) followed by Ringer's solution without CaCl₂. The coverslips were then exposed to hypotonic Ringer's solution (one part of calcium-free Ringer's solution was diluted in two parts deionized water) to swell the cells and facilitate easy opening[88]. To open the nuclei, a two-step procedure was developed. In the first step, cells were opened by placing an Alcian blue-coated coverslip on the cell-coated coverslip for ~1 min. After ~30 s, the excess buffer between the coverslips was wicked using a filter paper. After a further ~30 s, the coverslips were separated by a stream of phosphate-buffered saline (PBS, pH 7.4) (500–1000 μL) using a pipette. This facilitated the transfer of half-open cells with intact nuclei onto the top coverslip (Alcian blue-coated). The coverslips were then transferred to deionized water to swell the nuclei. The nuclei were then treated with Benzonase® nuclease (Merck) in PBS (supplemented with 2 mM Mg²⁺) to digest the chromatin, washed with a high-salt buffer (PBS with 300 mM NaCl, pH 7.4), and then re-equilibrated in PBS. In the second step, the nuclei were opened to expose the lamina. For this, again an Alcian blue-coated coverslip was placed on the coverslip with nuclei, the excess buffer removed using a filter paper, and the coverslips separated by a stream of paraformaldehyde (4%) or PBS for experiments with unfixed nuclei. Both the

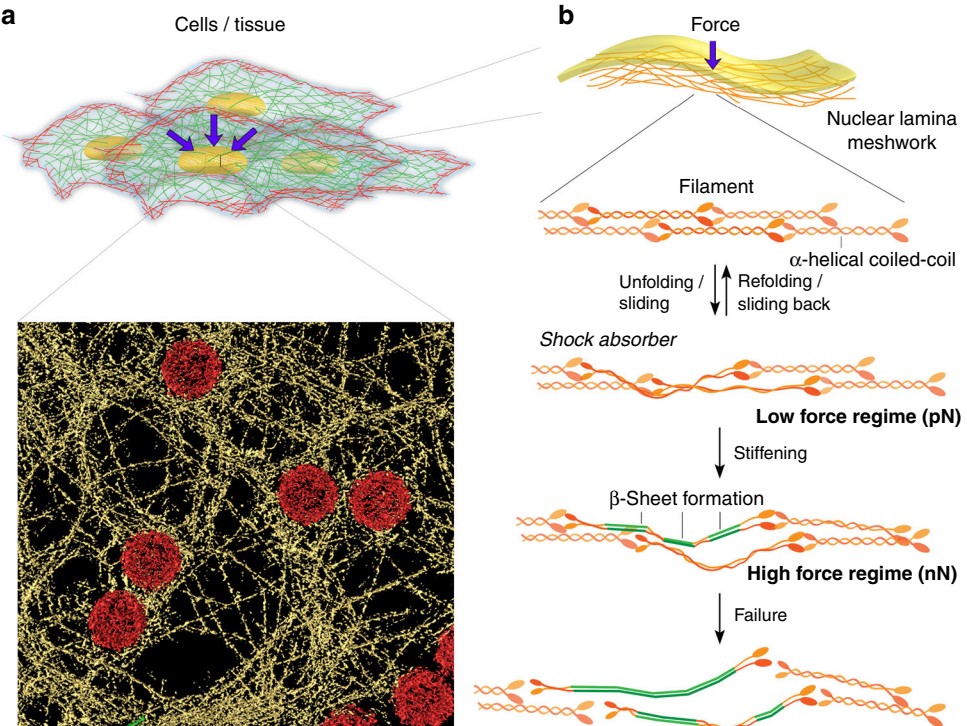

**Fig. 7 Nuclear lamins under external forces. a** The cell nucleus is under constant stress from its surroundings and experiences continuous or prolonged mechanical shocks during division and migration[93,94]. The nuclear lamina forms a meshwork that protects the genome and maintains the integrity of the nucleus. Surface-rendered view of the lamin meshwork in a MEF nucleus obtained by cryo-ET (adapted from ref. [4]). **b** A schematic model based on in vitro AFM measurements showing the response of lamin filaments when subjected to different levels of external force. Based on the experiments compressing the nuclei (Supplementary Fig. 20 and 21), we speculate that the meshwork topology probably plays a role in lamina mechanics. Whether the filaments undergo a similar fate during compression as observed in point load in vitro experiments, will require further investigation. Purple arrows denote loads exerted on a nucleus.

coverslips were screened for nucleus and the one with higher density used for imaging. The procedure was carried out with and without protease inhibitors (Merck) without any noticeable effect on the lamin meshwork.

**AFM imaging and force spectroscopy**. As explained above, oocyte nuclei were attached onto poly-L-lysine-coated glass and were mechanically opened to ensure that the nucleoplasmic side was facing up, i.e., accessible to the AFM cantilever (Fig. 1a, b)[87]. If the spread nuclear membrane folded (this happened usually at the edges because it could not stick to the poly-L-lysine surface), the sample was not measured in those regions. Since we always imaged and did our measurements in the center of the membrane, we are certain that all our measurements were conducted on the INM. In the AFM images of the lamin meshwork we took before pushing on the filament, no ruffles were observed. Moreover, folds and ruffles in the nuclear membrane can break the lamin meshwork.

Uncoated cantilevers of nominal tip radius ≈10 nm (HQ:CSC38/noAl, MikroMasch, Europe) were used for imaging and force spectroscopy. Before imaging and force spectroscopy, the cantilever sensitivity was determined by pressing the cantilever on a clean part (uncoated) of the glass surface. The spring constants measured using the in-built calibration module of the AFM (Nanowizard III, JPK instruments, Berlin) agreed with the typical range of the nominal spring constant of $0.03–0.13 \, \text{N m}^{-1}$. *X. laevis* oocyte nuclear lamina was imaged using AFM quantitative force imaging at $128 \times 128$ or $256 \times 256$ pixels. Random positions on lamin filaments were chosen in the closed-loop mode (feedback on); this is an entirely software-based "select and click" process that puts a cross-hair on the filament. Upon switching to the "force-spectroscopy" mode in the software, the tip of the AFM cantilever was pushed on the selected positions with a force of 8–10 nN at different velocities ($0.05–5 \, \mu\text{m s}^{-1}$). AFM experiments were performed in an acoustically isolated, temperature-controlled enclosure ($26 \pm 1 \, °\text{C}$). Importantly, the data were collected over a span of >2 years providing reproducible results ensuring that the quality and the selection procedure of the oocytes were maintained and did not affect the final results.

The FE signals were exported in ASCII format from the JPK analysis software (v 4.2). The parameters: step unit (distance between the peak and the linear drop in the peak), failure force (peak of the force signal), stiffness (slope of the linear steep increase in the force signal), and deformation (distance between the inflection in the FE signal and the first force peak) were all determined manually using Punias 3D

(v 1.0, Release 2.2)[89]. Specifically, the stifness before failure and after α-helix transition to β-sheet was determined by fitting a linear function to the relevant region of the curve—at the region prior to the peak occurrence (failure). The goodness of fit was estimated by the $R^2$ values (0.93–0.99) (Supplementary Fig. 9). More than 50% of the data was analyzed three times to ensure reproducibility of the analysis procedure. The results were always in agreement within <5%. The loading rates were determined as the product of the empirically measured stiffness (slope of individual force signals) and the respective speeds (0.05, 0.1, 0.2, 0.5, 1, 2, and $5 \, \mu\text{m s}^{-1}$).

In the MEF nucleus, the lamin meshwork is not in an orthogonal pattern (Fig. 7)[4]. Cryo-ET images from unswollen *X. laevis* oocyte nuclei also show that the native lamin meshwork is not always orthogonal (Supplementary Fig. 2). The elastic modulus (~25 mN $\text{m}^{-1}$) of the lamin meshwork was estimated to be similar in swollen and unswollen nuclei of *X. laevis* oocyte showing that the lamin meshwork is capable of large changes while maintaining its material properties[5]. We are therefore confident that in our open nucleus system, even if the meshwork was perturbed, the filament mechanics did not change drastically.

**Reversible pushing of lamin filament**. Sample preparation and positioning of the AFM cantilever tip on lamin filaments were performed as mentioned above. For the repetitive protocol, the cantilever was first pushed on chosen positions on lamin filaments with a force of 0.5 nN at a specific velocity ($0.1–2 \, \mu\text{m s}^{-1}$) and then retracted 1 μm from the filament at a specific velocity ($0.1–2 \, \mu\text{m s}^{-1}$). With a maximum retract velocity $2 \, \mu\text{m s}^{-1}$ of the cantilever over 1 μm, the refolding time of the α-helical coiled coil is estimated to be ~500 ms. The cantilever was pushed again on the same spot until the failure peak was detected.

**Nuclei purification for parallel-plate assay**. MEF cells were grown to 70–80% confluency in a T75 flask at $37 \, °\text{C}$ (5% $CO_2$ v/v), washed with 10 mL PBS and treated with trypsin for 3 min at $37 \, °\text{C}$. Reaction was stopped by resuspending the cells in medium containing FCS; the suspension centrifuged at $1000 \times g$ at $4 \, °\text{C}$ for 5 min. The cell pellet was washed with 5 mL cold ($4 \, °\text{C}$) hypotonic buffer (10 mM HEPES, 1 mM KCl, 1.5 mM $MgCl_2.6H_2O$)[90] and centrifuged at $1000 \times g$ for 5 min at $4 \, °\text{C}$. The pellet was incubated in 5 mL hypotonic buffer containing 0.1% w/v digitonin on ice for 20 min, dounced 26 times and centrifuged at $1000 \times g$ for 5 min at $4 \, °\text{C}$. The pellet containing nuclei was resuspended in 5 mL cold hypotonic

buffer, centrifuged ($1000 \times g$ for 5 min, 4 °C), pellet resuspended in PBS ($+2$ mM MgCl$_2$), and centrifuged again ($1000 \times g$ for 5 min, 4 °C). The purified nuclei were resuspended in 0.5–1 mL PBS ($+2$ mM MgCl$_2$) containing 1% w/v BSA.

**Parallel-plate assay for confining nucleus and measuring mechanical resistance.** Focussed ion beam-sculpted cantilevers were fixed on a standard JPK glass block and mounted in the AFM head (CellHesion 200; JPK Instruments). Nuclei were stained with NucBlue® reagent (ThermoFischer Scientific). The bottom surface of the nucleus (attached to the glass) was focused and the cantilever approached to touch the glass surface close to the nucleus; this was set as 0 µm. The cantilever was retracted 20–25 µm away from the surface, positioned above the nucleus, and approached to touch the nucleus. Both, the piezo height and the force experienced by the cantilever, were monitored simultaneously in two different channels (Supplementary Fig. 20). Using the two signals and the point of cantilever deflection, the height of the nucleus was determined (Supplementary Fig. 20). Cantilever calibration was carried out using the thermal noise method (in-built calibration module).

**Cryo-electron tomography (cryo-ET) of nuclear lamina.** *X. laevis* nuclear membranes were prepared in a similar manner as that for AFM experiments. Isolated nuclei were transferred onto a glow-discharged, perforated carbon copper grid (Quantifoil R2/1 200 mesh). The nuclei were opened manually and the NEs spread over the grid. The grid was washed three times with LSB to remove residual chromatin and oocyte debris. A 3 µL drop of BSA-conjugated 10 nm colloidal gold was applied, and the grid was vitrified by rapid plunge-freezing in liquid ethane. Tilt-series ($+60°$ to $-60°$) of the nuclear membrane was collected using Serial EM (v 3.8) in a Titan Krios microscope (ThermoFischer Scientific) equipped with an energy filter and a K2 Summit direct electron detector using the dose fractionation mode at 5 fps. The projection images were acquired at a defocus of 6 µm at ×42,000 magnification, corresponding to a pixel size of 0.34 nm. The sample was exposed to a total dose of ≈52 $e^-$ Å$^{-2}$. The tomograms were reconstructed using TOM Toolbox and rendered in Amira software (v 6.0, Thermo Fisher Scientific). Tomograms of MEF nuclear lamina were acquired and rendered following published procedures[4].

**SDS-PAGE and western blot analysis.** Cells (WT, lamin A ko, lamin B dko) were harvested, counted with a hemocytometer and lysed in an appropriate amount of SDS sample buffer ($4 \times 10^4$ cells/10 µL sample buffer). Ten microlitres of each sample was loaded on a precast gradient gel (4–12%) and proteins were separated by SDS-PAGE according to their molecular weights. Next, the separated proteins were blotted on a PVDF membrane (1.1 mA cm$^{-2}$ for 90 min). Afterwards, the membrane was blocked with 5% dry milk in PBS-T (PBS, 0.1% Tween® 20) for 30 min at RT and then incubated with primary antibody (Abcam, anti-lamin A/C, EP5420, ab133256, 1:1000), anti-lamin B1 antibody (Santa Cruz, Lamin B1 (B-10): sc-374015, 1:500), diluted in 5% dry milk in PBS-T for 1 h to overnight at RT or 4 °C, respectively. The membrane was washed $3 \times 10$ min with 5% dry milk in PBS-T and incubated with secondary antibody (HRP-labeled donkey anti-rabbit, -mouse, or -goat antibody) diluted in 5% dry milk in PBS-T for 1 h at RT. Then, the membrane was washed $3 \times 10$ min with 5% dry milk in PBS-T and once with PBS. To visualize the secondary antibody, Immobilon Western Chemiluminescent HRP substrate (Millipore) was added and the signal was detected using a Fujifilm LAS-3000 Imaging system.

**Meshwork analysis.** Lamin filament density in individual slices of the tomograms were rendered manually using the Amira software. Thirteen tomograms from *X. laevis* nuclei and 12 from MEFs nuclei were analyzed and rendered. The rendered 3D tomograms were automatically converted into graphs using Amira. The vertices and links were then manually checked to verify the automated procedure. Closed loops, overlying links and extra vertices were manually deleted upon comparison with individual slices of the tomograms. The graph coordinates were then exported to Cytoscape (v 3.4.0) and the parameters analyzed using the network analyzer module.

**MD simulations.** *Scale-free model of the lamin meshwork*: The mesoscopic model of the lamin meshwork used here is based on a combination of experimental and full atomistic data. The geometry of the meshwork was obtained from our experimental observations that give its topological feature to be scale free. The meshwork was initially generated by deciding the connectivity of the end nodes of all the lamin filaments with the node list and their connectivities built according to a Barabasi–Albert model. This connected node list was further modified by repeating a simple Monte-Carlo process for 1,000,000 times to randomly select a node and change its connectivity. This change will be more likely to retain if it makes the degree probability distribution closer to the desired Power-law distribution ($P(k) \sim k^{-\lambda}$) with the desired scaling factor $\lambda$. We randomly assigned the coordinates of all the nodes in a two-dimensional plane that mimic the effect of nuclear membrane and treat each connection between the nodes as an existing lamin filament. Thereafter, we ran another simple Monte-Carlo process for 600,000 times and for each one we took the coordinates of a randomly selected node to have a random move. By doing so we ensured that the filaments in the meshwork

have an average length of $12 \pm 3$ nm. The entire process was repeated to obtain meshwork models with different $\lambda$ values. To account for the random effects introduced in the model generation, we performed individual loading tests on 100 meshworks and statistically studied the mechanical response from simulations.

*Multiscale-modeling of lamin filament*: To address the failure mechanism of the filament meshwork, the interpretation of the dynamic fracture property of the entire meshwork of complex topology requires a large-scale meshwork model as well as accurate representation of the dynamical property of each filament and their interactions with atomic interaction detail (in the order of Å). Thereby, the multiscale modeling can be a strategy combining accuracy and efficiency, which includes the full atomic modeling for single filaments and the mesoscopic model for IF meshwork. The full atomic model provides an accurate description of the physical mechanisms governing the yielding (unfolding of alpha-helics) and stiffening (structural transition) process of each filament under loading, and records the accurate description of the FE relation of the single filament during the loading history[52]. The atomic simulations were carried out using the CHARMM19 all-atom potential energy function with an effective Gaussian model for the water solvent[91]. We apply a constant temperature (300 K) controlled by a Berendsen thermostat. The alpha-carbon atoms at the two ends of a tetramer are pulled on by using steered MD with a pulling velocity of 0.1 Å ps$^{-1}$ and a stiffness of 10 kcal mol$^{-1}$ Å$^{-2}$ using the CHARMM simulation package.

Simulation results in the current paper are obtained from a mesoscopic model, which inherits the physical properties of the atomic ones and provides a faithful estimation of dynamic fracture property in the large scale. We used mesoscopic beads to model each filament within the meshwork and fix $r_0 = 1$ nm as the initial distance between neighboring beads as well as the equilibrium length of the springs. This length is much smaller than the persistence length of the lamin filament (≈1 µm). Therefore, each filament is modeled by a series of beads interacting according to nonlinear inter-particle multibody potentials (as the force field for the mesoscopic model) that is obtained from full atomistic simulations[32,52]. This force field of our mesoscopic system is given by[32]

$$E_x = E_T + E_B + E_{nonbond}, \quad (1)$$

where $E_T = \sum_{pair} \varphi_T(r)$ is the energy term that associates with the tensile deformation of the IF fiber, $E_B = \sum_{triplets} \varphi_B(\theta)$ is the energy term for the bending energy and $E_{nonbond} = \sum_{pair} \varphi_{nonbond}(r)$ is the non-bonded energy term for two fibers in contact. We approximated the nonlinear FE behavior of a single fiber under tensile loading with a multi-polynomial potential model[32,52] and the tensile force between two bonded neighboring beads is given by

$$F_T(r) = \left[ \exp\left(\frac{r - r_b}{r_b} \Xi\right) + 1 \right]^{-1} \begin{cases} k_1(r - r_0) & r < r_1 \\ R_1 + k_2(r - r_1) & r < r_2 \\ R_2 + k_3^1(r - r_2) + k_3^2(r - r_2)^2 + k_3^3(r - r_2)^3 & r < r_3 \\ R_3 & r \geq r_3. \end{cases} \quad (2)$$

In Eq. (2), $r$ is the bond length and $k_i$ and $r_i$ are spring constants that derived directly from the FE curve of the tension test of full atomic model[32,52], their value are given as $k_1 = 0.7975$ kcal mol$^{-1}$ Å$^{-2}$, $k_2 = 0.162$ kcal mol$^{-1}$ Å$^{-2}$, $k_3^1 = 1.022$ kcal mol$^{-1}$ Å$^{-2}$, $k_3^2 = 0.365$ kcal mol$^{-1}$ Å$^{-3}$, $k_3^3 = 0.116$ kcal mol$^{-1}$ Å$^{-4}$, $r_0 = 10$ Å, $r_1 = 15$ Å, $r_2 = 19$ Å, and $r_3 = 27.6$ Å. The force continuity conditions $R_1 = 3.988$ kcal mol$^{-1}$ Å$^{-1}$, $R_2 = 4.635$ kcal mol$^{-1}$ Å$^{-1}$, and $R_3 = 113.936$. The Fermi–Dirac distribution function introduces two additional parameters $r_b = 36$ Å and $\Xi = 300$. The parameter $r_b$ denotes the critical separation distance for breaking of the filament and the parameter $\Xi$ describes the amount of smoothing around the breaking point (the smaller $\Xi$, the smoother the curve becos). Similar strategy to model the potential near rupture for numerical analyses has been done in earlier work. The bending energy is given by

$$\varphi_B(\theta) = \frac{1}{2} k_B(\theta - \theta_0)^2, \quad (3)$$

with $k_B = 34.32$ kcal mol$^{-1}$ rad$^{-2}$ relating to the bending stiffness of the IF *EI*. The non-bonded interaction $\varphi_{nonbond} = A\left[1 + \cos\left(\frac{\pi r}{r_c}\right)\right]$ with $r_c = 10$Å and $A = 4$ kcal mol$^{-1}$ that prevents two filaments to overlap and penetrate during the simulation.

*Computational experiments on the meshwork*: Calculations were carried out in two steps: (1) relaxation followed by (2) loading. We modeled the effect of the nuclear membrane as a fixed plane substrate that has van der Waals interactions with the lamin filaments with a surface energy of 20 mJ m$^{-2}$ at an equilibrium distance of 1.5 nm and cut-off distance of 4.0 nm. This energy is in the same order as vimentin protein adhesion on a silica surface and cell adhesion on a mineral surface[92]. Relaxation was achieved by heating up the system, then annealing the structure at a temperature of 300 K, followed by energy minimization. After relaxation, the system was maintained at 300 K in an NVT ensemble (constant temperature, constant volume and constant number of particles) and loading applied by displacing a point within a single filament (by applying a point loading in the normal direction with the meshwork). While the single point was under a pushing force, the rest of the beads within the cut-off distance interact with the substrate, giving the reaction force to deform the filament. This set-up resembles

the AFM pushing at a single point on a lamin filament, continuously displacing particles in the boundary at a speed of 0.1 mm s$^{-1}$. It is confirmed that the loading rate chosen here is slow enough that leads to quasi-static deformation conditions. Each of the computational simulations reported in this paper was carried out using the LAMMPS simulation package. Other steps, including network generation and post-analysis for statistical calculations, were performed by using customized scripts in MATLAB. More details on the modeling of each filament and numerical values of the physical terms can be found in previous papers[32,52].

**Reporting summary**. Further information on research design is available in the Nature Research Reporting Summary linked to this article.

## Data availability
Data supporting the findings of this study are available from the corresponding authors upon reasonable request. Source data are provided with the paper.

## Code availability
Codes, algorithms, associated protocols, and scripts are available upon request.

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

## Acknowledgements

The authors thank Y. Turgay for discussions and providing tomograms of MEFs nuclear lamina and R. Tenga for the Western blot. K.T.S. appreciates support from Forschungskredit fellowship, University of Zurich. This work was funded by a Swiss National Science Foundation Grant (SNSF 31003A_179418), the Mäxi Foundation to O.M. Additional support was provided by the Office of Naval Research (N00014-16-1-2333) to Z.Q. and M.J.B. We thank the Center for Microscopy and Image Analysis at the University of Zurich.

## Author contributions

K.T.S. and O.M. conceived the study concept with input from U.A. K.T.S. developed the idea of combining mechanics and cryo-ET with MD simulations, performed the AFM experiments and analyzed the AFM and cryo-ET data. A.D.-G. performed the cryo-ET experiments on frog oocyte nuclei. Z.Q. performed the simulations. Z.Q. and M.J.B. analyzed the data. D.J.M. advised on the parallel-plate assay. K.T.S. wrote the paper with input from Z.Q. on MD simulations. All authors contributed to the final writing of the paper.

## Competing interests

The authors declare no competing interests.
