## [Peer Review File · Nature Communications]

Reviewers' comments:

Reviewer #1 (Remarks to the Author):

In this well-written manuscript, Medalia and colleagues present an interesting study on lamin mechanics. They used AFM in combination with cryo-electron tomography and molecular dynamic simulations to explore lamin organization and its mechanical response to force. While this work provides an interesting set of data concerning lamin mechanics, it is mainly descriptive and only offers hypothetical explanations (mostly based on simulations) regarding the molecular mechanisms involved.

Adequate control experiments have been conducted to look at the response without filaments, however other experiments (which one could consider as control) may have helped to confirm the potential explanations offered by the authors. For example, whereas the step sizes (1.3 and 4 nm) are coherent with α helix coiled coil rupture and failure of stiffened tetramers respectively, the authors can't rule out other potential mechanisms (such as interprotein sliding or changes in lamin interaction with the INM *via* INM proteins). Using their main model (Xenopus oocyte) or their alternative model (MEF) the authors could have performed a combination of depletion and re-expression (of lamin mutants) experiments to confirm their hypothesis (or force clamp experiments on purified lamin filaments).

Some parts would benefit from additional explanations, such as line 126-127 and line 142, where it is unclear what evidence from Supp. figure 3 (different size of probes) or from previous publications (ref 30 and 32) are used by the authors to conclude. It could be interesting to have a part of Supp Figure 3 d on Figure 1 (after d) to compare *in situ* and *in silico* data. It is unclear whether the authors performed experiments on MEFs (line 207-209) or adapted results from previous publication (ref 24) as suggested by Fig 5 legend (line 603)?

Other comments:

-Fig 5. The main experimental model (Xenopus laevis oocyte) may contain only lamin III (B type) and MEF have additional types of lamins, the authors did not comment these differences in regards to the meshwork analysis.

-Some figures are lacking explanations, such as Figure 4 c (green curve?)

-Line 197. Did the authors mean α -helix to β -sheet stiffening or α -helix to β -sheet transition?

Reviewer #2 (Remarks to the Author):

Title: Nonlinear mechanics of lamin filaments and the meshwork topology build a hierarchical nuclear lamina

General comments:

This could be interesting work. However, the way it is presented here, it is almost impossible for a reader to glean any benefit from the authors work. For this to be evaluated and, in my opinion, to

be of use to readers, it has to be completely re-written to clarify the grounds for the authors claims. They may have very good grounds for making the statements they do about the experiments. However, it is not obvious from what is presented here.

Page 3: “Nuclei were opened manually, chromatin removed and NEs spread on a poly-L-lysine coated glass surface such that the lamin meshwork was accessible to the AFM cantilever tip. The lamina was imaged by FE-based imaging (FE-imaging), recording FE curves at specified pixels of the sample at a force of 0.5 – 0.75 nN in the closed-loop. As observed previously by EM24-26, lamin filaments were arranged...”

I have a few concerns about the experimental system. I would help if the authors could address these concerns which address the question of the extent to which the in vitro system that they query is quantitatively recapitulating the behavior in the intact cell.

First, how do they know that the lamin, when removed from the cell, behave anything like lamin in the contact of the cell? Is it not possible, that upon removal from the cell, many proteins, small molecules, dissociate from the lamin which radically alter their response properties to loads? Even a change of ionic milieu or colloid-osmotic strength could have a significant effect.

Second, how do they know that anchoring various domains to the lysine coating cover slips does not radically change the ability of the lamins to respond to load?

Third, they are pressing on a complex network of polymers. If I understand the experiments correctly, they do not have a specific molecular interaction between their probe and specific polymers, much less specific locations on the polymers. In the absence of this information, it is not clear to me how they are making their conclusions.

Page 4:

“Previous MD simulations of stretching vimentin²⁷ and lamins, in an orthogonal meshwork¹⁹ (Supplementary Fig. 4), also showed nonlinear stress-strain profiles and failure forces similar to those observed by mechanical pushing of lamin filaments in situ (Fig. 1d).”

In the text it says that supplemental figure 4 is an MD simulation, but this is not mentioned at all in the figure legend. Is it an MD simulation or not? How are they similar or different? It is clear that the authors have thought about this and are trying to make a point. It would help to share this insight with the readers. From the two curves it is difficult to judge the significance of the similarity. Isn't this kind of pattern seen with many different biological molecules?

Page 5:

“Similarities with the in vitro and in silico FE curves indicate that we probed single lamin filaments in situ.”

What would the curve look like if it were two lamin filaments? Five lamin filaments? There is no calibration presented from which a reader can judge the conclusions.

Page 5:

“The plateau presumably denotes unfolding and clamping of an α -helical coiled-coil, preceding its conversion into β -sheets, and the intermediate peak denotes the failure of dimer-dimer interactions”.

What is the reason for concluding that these steps are presumably anything? I am sure that there

are some grounds for this, but it was not make clear to the reader.

“In the high force regime beyond the yield point (Fig. 1d), lamin filaments showed plastic deformation.”

How is it known that at the yield point it is the lamin filaments, not the nuclear filaments, the nuclear membrane of the nucleoplasmic nuclear basket. Given that 100% of the membrane is not going to be nucleoplasmic side out (these membranes fold back on themselves when the nuclei are opened), maybe it is the outer nuclear membrane or maybe sheets of endoplasmic reticulum that are continuous with the outer nuclear membrane. How is the orientation of the nuclear membrane on the surface determined both globally, and very locally where they are probing?.

Page 5:

“Interestingly, $\approx 60\%$ of the FE curves also showed a second peak with a step unit of ≈ 4 nm.”

But this is a correlation, and a weak one at best. There are many things in the cell at 4 nm including proteins and the thickness of the membrane.

Page 5:

“The outer and inner nuclear membranes of frog oocytes are ≈ 50 nm apart, while NPCs are ≈ 90 nm tall structures³⁴ with flexible lamin filaments situated on the cytoplasmic face. This enabled pushing of lamin filaments up to 100 nm towards the glass surface.”

I do not understand how the authors reached this conclusion.

Page 6:

“To answer this, we applied a repetitive force protocol to measure the energy dissipated during filament failure. Independent of the loading rate, the energy dissipated in the low force regime was determined to be ≈ 10 -17 J (Fig. 3a, c), increasing to ≈ 10 -16 J (105 kBT) up to the apparent failure (Fig. 3b, d). Majority (89%) of the FE curves ($n = 133$) showed reversibility of the plateau up to 0.5 nN indicative of fast refolding (< 50 ms) of the α -helical coiled-coil domain.”

It is not clear to me what the logic was of the experiments or how the authors reached this conclusion.

Page 7:

“The similar values of the step sizes as in constant velocity experiments indicate that the 1.3 nm steps signify the failure of single α -helical coiled-coils in a filament, and the 4 nm steps at ≥ 2 nN denote the failure of tetramers.”

This seems like a leap of faith. Why assume it is from the lamins?

“The lifetime, τ break, of the α -helical coiled-coils at $F_{load} \leq 1$ nN was measured to be a few hundred milliseconds, suggesting that the coiled-coil structure is the first force buffer. As the force is increased to 3 nN, the α -helix to β -sheet stiffening increases the load bearing capacity of the filament, and the failure requires tens of milliseconds (Supplementary Fig. 10).”

How, based on the information shown here, have they assigned the force to the lamins, and then how have they assigned which force is to the alpha helices and which to the beta sheet?

“Interestingly, both, *X. laevis* oocyte and the mammalian nuclear lamin meshworks, exhibit similar

topological features of the laminae.”

What are the criteria for calling them “similar”. The slopes of c and g in figure 5 look different to me. Is there a statistical analysis they can use? Can they examine other features of the cell to give a sense of what is similar and what is different? I find this sentence and the data, as presented, as uninformative. What if a similar analysis was applied to microtubules or actin under the membrane?

“and points to the importance of hubs as important signal transmission points in the meshwork.”

Where is any hint that these are involved as signaling? This sentence comes completely out of context.

Point by point answers to Reviewers' comments:

Reviewer #1 (Remarks to the Author):

Reviewer: In this well-written manuscript, Medalia and colleagues present an interesting study on lamin mechanics. They used AFM in combination with cryo-electron tomography and molecular dynamic simulations to explore lamin organization and its mechanical response to force. While this work provides an interesting set of data concerning lamin mechanics, it is mainly descriptive and only offers hypothetical explanations (mostly based on simulations) regarding the molecular mechanisms involved.

We thank the reviewer for the constructive comments. We have addressed the concerns of the reviewer, included new data, which has made the paper more focused and conclusive.

Adequate control experiments have been conducted to look at the response without filaments, however other experiments (which one could consider as control) may have helped to confirm the potential explanations offered by the authors. For example, whereas the step sizes (1.3 and 4 nm) are coherent with α -helix coiled coil rupture and failure of stiffened tetramers respectively, the authors can't rule out other potential mechanisms (such as interprotein sliding or changes in lamin interaction with the INM *via* INM proteins).

The reviewer is correct in pointing out that there could be other potential mechanisms. Based on MD simulations and previous experiments on other filament proteins (Block et al., 2018; Makarov et al., 2019) it is possible that the filaments slide slightly before the unfolding and the α -helix to β -sheet transition. When the lamin meshwork is stretched (**Supplementary Fig. 7**), there is sliding at high forces (nanoNewton) during filament failure. In the present study, MD simulations of pushing on single lamin filaments in a meshwork did not show filament sliding in the low force plateau region or the high force stiffening region at failure (**Supplementary Fig. 6**). Sliding may occur more frequently at lower speeds (as in experiments) as at higher speeds ($\gg 1 \mu\text{m s}^{-1}$) sliding requires much larger forces than unfolding coiled-coil α -helices thereby making unfolding the dominant event. MD simulations are performed at a few orders of magnitude higher speeds than experiments, and could be a reason for not observing clear sliding. In line with the reviewer comments, we have mentioned this possibility in the revised manuscript (**page 6**).

Moreover, as in MD simulations where a plateau is observed at lower forces (300 – 500 pN) and attributed to unfolding / sliding of filaments, force-extension (FE) curves from AFM experiments also showed a plateau at similar forces (as shown in **Figures 2a, b**). The intermediate and final step sizes therefore are not attributed to the sliding but to breakage of molecular interactions.

We thank the reviewer for raising the contribution of INM proteins to the lamins and lamina organization. Of course, the NE from frog contains INM proteins that presumably interact with the LIII lamin. These interactions may serve as an additional anchor of the lamins to the INM (in addition to the farnesyl groups at the C-terminal of lamin) and may potentially reduce sliding of the lamin filaments. The inter-connectedness of the filaments at junctions may also resist sliding and promote strain-induced stiffening. Since the measurements were done on thousands of positions along lamin filaments, the force measured here is of lamins. Nevertheless, we have now made it clear that INM proteins may contribute to the low force region (**page 8**).

Using their main model (*Xenopus* oocyte) or their alternative model (MEF) the authors could have performed a combination of depletion and re-expression (of lamin mutants) experiments to confirm their hypothesis (or force clamp experiments on purified lamin filaments).

We agree with the reviewer that depletion / mutations are definitely interesting ideas and conducted new experiments as suggested (see below).

As already mentioned in **Methods**, we used stage VI nuclei from *Xenopus laevis* (*X. laevis*) oocyte for our experiments. The lamin meshwork of these nuclei are predominantly composed of LIII lamins (Aebi et al., 1986). Protein depletion in this experimental system would require the production of transgenic frog. The focus of our study was to characterize the mechanical properties of wild-type lamin filaments assembled *in situ*. This is a major novelty, and would allow to explain the resistance of nuclei to external forces. Further, it is important to note that vertebrate lamins cannot be reconstituted to form stable lamin filaments *in vitro*, but rather assemble into less physiologically relevant paracrystalline arrays. Moreover, lamin filaments cannot be purified to retain their native structure, and recombinant expression of vertebrate lamins produces paracrystals rather than stable filaments (Moir et al., 1991; Stuurman et al., 1998; Turgay et al., 2017). Therefore, *in vitro* analysis of vertebrate lamin mature filaments is not yet possible.

However, to confirm the hypothesis that the meshwork topology affects mechanical properties of the nucleus, **we conducted new experiments using** mouse embryonic fibroblasts that express a single lamin isoform (lamin A or lamin B) (**Supplementary Figs. 19, 20**). Although it is not the main experimental model of this work, it provides additional support that altering lamin type affects mechanics, likely due to changes in the meshwork topology and filaments interactions.

Some parts would benefit from additional explanations, such as line 126-127 and line 142, where it is unclear what evidence from Supp. figure 3 (different size of probes) or from previous publications (ref 30 and 32) are used by the authors to conclude.

We apologize for the confusion. The simulation experiments shown in the previous **Supplementary Fig. 3** (now **Supplementary Fig. 6**) were done with different radii of cantilever probe to examine and better understand the AFM experimental results. The idea was to check if the tip curvature has an influence on the filament mechanics. We have removed the citations to avoid confusion as the simulations are new. Additionally, we have updated the revised manuscript (**page 6**).

It could be interesting to have a part of Supp Figure 3 d on Figure 1 (after d) to compare *in situ* and *in silico* data.

We thank the reviewer for the excellent suggestion. We have modified **Figure 1** accordingly.

It is unclear whether the authors performed experiments on MEFs (line 207-209) or adapted results from previous publication (ref 24) as suggested by Fig 5 legend (line 603)?

We thank the reviewer for pointing out this source of confusion. We have now clarified this in the revised text (**pages 10/11**), in line with the legend of **Figure 5**. Data acquisition was

previously reported (ref 24) but the entire meshwork analysis is novel and presented in this manuscript.

Other comments:

-Fig 5. The main experimental model (*Xenopus laevis* oocyte) may contain only lamin III (B type) and MEF have additional types of lamins, the authors did not comment these differences in regards to the meshwork analysis.

We thank the reviewer for raising this important point. Surprisingly, despite the clear differences between the molecular constituents of the lamina, network analysis suggests that lamins are not randomly arranged at the lamina. This statement is based on meshwork topology features that are common to both *X. laevis* and MEFs. We have added few sentences in the **Results** section (Meshwork topology influences lamin mechanics – **page 11**) in order to emphasize the **similarities and clear differences**. Indeed it is an important finding of the present work.

The meshwork model used for MD simulations is a physical model based on data acquired by our cryo-ET experiments on nuclei from *X. laevis* oocytes and MEF cells.

-Some figures are lacking explanations, such as Figure 4 c (green curve?)

We thank the reviewer for pointing out on this error. We have now removed the green curves from figures with histograms in the revised manuscript.

-Line 197. Did the authors mean α -helix to β -sheet stiffening or α -helix to β -sheet transition?

We agree with the reviewer that this part was confusing. We meant α -helix to β -sheet transition, and the β -sheet is stiffer than α -helix. We have now revised this statement to avoid ambiguity (**page 10**).

Reviewer #2 (Remarks to the Author):

General comments:

This could be interesting work. However, the way it is presented here, it is almost impossible for a reader to glean any benefit from the authors work. For this to be evaluated and, in my opinion, to be of use to readers, it has to be completely re-written to clarify the grounds for the authors claims. They may have very good grounds for making the statements they do about the experiments. However, it is not obvious from what is presented here.

We thank the reviewer for appreciating the importance of this work. We have revised the manuscript, modified and added statements emphasizing the biological implications of our results. Additionally, we have tried to explain better the physical experiments and the molecular dynamics simulations.

Page 3: “Nuclei were opened manually, chromatin removed and NEs spread on a poly-L-lysine coated glass surface such that the lamin meshwork was accessible to the AFM cantilever tip. The lamina was imaged by FE-based imaging (FE-imaging), recording FE curves at specified pixels of the sample at a force of 0.5 – 0.75 nN in the closed-loop. As observed previously by EM24-26, lamin filaments were arranged...” I have a few concerns about the experimental system. It would help if the authors could address these concerns which address the question of the extent to which the in vitro system that they query is quantitatively recapitulating the behavior in the intact cell.

First, how do they know that the lamin, when removed from the cell, behave anything like lamin in the contact of the cell? Is it not possible, that upon removal from the cell, many proteins, small molecules, dissociate from the lamin which radically alter their response properties to loads? Even a change of ionic milieu or colloid-osmotic strength could have a significant effect.

We thank the reviewer for raising these concerns. Several studies have been previously conducted where nuclear mechanics were determined inside the cell and on purified nuclei. It was shown by Rowat *et al.* that nuclear mechanics is the same inside the cell and when removed from the cell (Rowat *et al.*, 2005). In the case of *X. laevis* oocyte nuclei, similar elastic moduli ($\sim 25 \text{ mN m}^{-1}$) of lamin meshwork were determined in swollen and un-swollen nuclei showing that the lamin meshwork maintains its properties in spite of large changes in the nucleus (Dahl *et al.*, 2004). This is now mentioned in **Methods (pages 16/17)**.

The intimate interactions between lamins and chromatin in mammalian cells prevent unambiguous determination of the mechanical properties of lamins. In any case, and to further address the reviewer’s concerns, we have performed experiments to open ‘windows’ in the nuclei of MEFs and imaged lamin filaments (see **Supplementary Fig. 1**). The new additional experiments on opening mammalian nuclei by de-roofing showed that when chromatin was not digested by Benzonase nuclease, lamin filaments were not observed clearly. This hindered using the sample for mechanical characterization of lamins with specificity (mentioned on **page 4**). Therefore, in this study we focused on *X. laevis* oocytes developed to stage VI in which the chromatin is condensed in the center of the nucleus and does not interact with the lamina. The oocyte nucleus was opened with a sharp needle, the nuclear envelope spread manually on poly-

L-lysine coated glass surface and the chromatin was washed off with buffer. For mechanical measurements of lamin filaments, the system provided closest-to-*in situ* view of the lamina (first studied by (Aebi et al., 1986)), containing unperturbed lamin filaments (Stanley et al., 2018; Stick and Goldberg, 2010). We have mentioned this now in the revised text (**Results, pages 4, 5**) and in the **Methods** section.

Nuclei purification exhibits no detectable alteration of lamin structures, as previously reported (Shimi et al., 2015; Turgay et al., 2017). As mentioned above, we utilized the *X. laevis* oocyte system because chromatin is not attached to the lamins. Thus, by scratching the nucleus with a sharp micro-needle spontaneous detachment of chromatin that can be gently washed and removed completely. Our lamin experiments are performed as described in the literature over the past decades. We first used a low salt buffer to swell the *X. laevis* oocytes to remove the nuclei, transfer the nuclei to Modified Barth's solution. This is already mentioned in the manuscript. In our hands, changing the buffers (e.g., Modified Barth's solution into PBS or HEPES) had no effect on the structure of the nuclear lamina of *X. laevis* oocytes as judged by cryo-ET (also documented in (Grossman et al., 2012)).

We have added citations to clarify this procedure. Since each LIII lamin filament is physically anchored to the nuclear envelope via farnesyl groups, B-type lamins are attached to the nuclear membrane. Therefore, majority of the lamin filaments are unperturbed by the procedure used here. A sentence clarifying this is added on **page 5**.

The reviewer is correct that ions and lamin-associated proteins may be removed from the nucleus by the procedure used here. Therefore, we only referred to the contribution of lamin filaments to the load and do not discuss the contributions of other components that may have been lost (e.g., DNA). Here, our aim was to measure the physical properties of the nuclear lamin filament in a system of reduced complexity, using a bottom-up approach. In a bottom-up approach the idea is to understand the behavior of individual components separately and then of the entire system and develop models of how the contribution of the individual components to the entire system. To achieve this, we probed lamin mechanics in a very simplified context while maintaining the meshwork and the nuclear membranes in a close to native state. This has never been attempted before and is the crux of the paper.

Ionic strength is known to have an effect on chromatin packing and binding. This is especially true for mammalian cells where chromatin is tightly attached to the inner nuclear membrane. Chromatin is however removed from our *X. laevis* oocyte nuclei samples before measurements. To the best of our knowledge, there are no studies showing that ionic strength changes the mechanical properties of lamin filaments. However, to address this, we have conducted control experiments using nuclease treatments to be absolutely certain of the effects of nucleic acids and found no difference compared to experiments done without nuclease treatment. This is a new figure (**Supplementary Fig. 14**).

On a side note, we would like to mention that historically *in vitro* analysis of macromolecular assemblies has been very informative and has pushed biology forward (examples can be cited from structural biology, biochemistry and even cell imaging that often uses detergents to extract cellular components). With this perspective, we actually used a minor purification approach (*i.e.*,

no detergents, centrifugations or other destructive steps) that kept lamins filaments organized in a meshwork and attached to the native inner nuclear membrane.

Second, how do they know that anchoring various domains to the lysine coating cover slips does not radically change the ability of the lamins to respond to load?

We apologize that the experimental set-up was not clearly explained. Neither the lamins nor the INM are in contact with poly-L-lysine coated glass surface. Therefore, no interactions between lamin domains and lysine are possible. We have added a sentence on **page 5** to clarify this and a schematic illustration (**Supplementary Fig. 3**).

Third, they are pressing on a complex network of polymers. If I understand the experiments correctly, they do not have a specific molecular interaction between their probe and specific polymers, much less specific locations on the polymers. In the absence of this information, it is not clear to me how they are making their conclusions.

The reviewer is correct that specific attachment between a cantilever tip and protein is sometimes used (although not always) to pull proteins for unfolding / folding. Here, we pushed with the cantilever tip and did not rely on any specific interactions between the cantilever tip and the filament. Previous experiments of pushing microtubules (Schaap et al., 2006), intermediate filaments adsorbed on a holey surface (Guzmán et al., 2006) and pushing intermediate filaments along a surface (Kreplak et al., 2005; Kreplak et al., 2008) did not use functionalized cantilevers to establish specific interactions between the cantilever tip and the filaments. Similar to previous studies on different materials and proteins, one can learn about the physical properties of lamin filaments (an intermediate filament protein) in the present work.

Our conclusions are based on the experimental observations supported by several controls (see **Supplementary Figs. 4, 5, 14**) and MD simulations. MD simulations, both in this and a previous work (Qin and Buehler, 2011), predict the behavior of pushing lamin filaments that was observed here experimentally. These were already discussed in the text.

Page 4: “Previous MD simulations of stretching vimentin²⁷ and lamins, in an orthogonal meshwork¹⁹ (Supplementary Fig. 4), also showed nonlinear stress-strain profiles and failure forces similar to those observed by mechanical pushing of lamin filaments in situ (Fig. 1d).” In the text it says that supplemental figure 4 is an MD simulation, but this is not mentioned at all in the figure legend. Is it an MD simulation or not? How are they similar or different? It is clear that the authors have thought about this and are trying to make a point. It would help to share this insight with the readers. From the two curves it is difficult to judge the significance of the similarity. Isn't this kind of pattern seen with many different biological molecules?

We apologize for the confusing text. **Supplementary Fig. 4** (now **Supplementary Fig. 7** in the revised manuscript) is an MD simulation; we have now mentioned this in the caption of the revised Supplementary information. As the reviewer correctly noticed **Supplementary Fig. 6** (previously **Supplementary Fig. 3**) and **Supplementary Fig. 7** (previously **Supplementary Fig. 4**) are different. In MD simulations of **Supplementary Fig. 6**, the lamin filaments were subjected to force perpendicular to the long axis of the filament as in the AFM experiment. In MD simulations of **Supplementary Fig. 7**, the lamin meshwork was stretched in its plane. We

have now clarified these points in the text of the revised Supplementary information. The structural intermediates observed in both simulations were the same except the sliding of β -sheets at failure when the meshwork was stretched (**Supplementary Fig. 7**).

The response of *in vitro* assembled vimentin when stretched at low forces resembles the response of lamins although vimentin filaments are much thicker. We are referring to this work in the new version (Block et al., 2018). Fibrinogen, fibrin and other IFs also show similar mechanical response. Those studies are cited (**page 7**).

Page 5: “Similarities with the *in vitro* and *in silico* FE curves indicate that we probed single lamin filaments *in situ*.”

What would the curve look like if it were two lamin filaments? Five lamin filaments? There is no calibration presented from which a reader can judge the conclusions.

We thank the reviewer for these questions. The MD simulations were performed on pushing single filaments in a realistic meshwork model based on real cryo-ET data of *X. laevis* oocyte and MEF nuclei. The FE profile *in silico* highly resembled the FE profile measured in the AFM experiments (*in situ*). These observations suggest that a single lamin filament was probed in the AFM experiments. Hence, we believe the statement is justified.

In our experiments we typically pushed on a single lamin filament, as confirmed by AFM imaging. Additionally, cryo-ET images (**Supplementary Fig. 8**) clearly show that in most cases distances between neighboring filaments are typically larger than the thickness of the AFM tip (<10nm). However, close interactions between two filaments are rarely seen. Therefore, to address what would happen if more than a single filament fails one could hypothesize that an increase in peaks in the FE curves should be observed. Indeed, in rare cases, we observed two or more peaks (max 5) in the FE curves in the AFM experiments. However, the occurrence probability of the peaks decreased with the number of peaks. We have analyzed of the occurrence probability of those peaks and added a new figure (**Supplementary Fig. 13**). These results are mentioned in the revised version (**page 8**).

Page 5: “The plateau presumably denotes unfolding and clamping of an α -helical coiled-coil, preceding its conversion into β -sheets, and the intermediate peak denotes the failure of dimer-dimer interactions”. What is the reason for concluding that these steps are presumably anything? I am sure that there are some grounds for this, but it was not make clear to the reader.

The reversible low-force alteration of various filaments has been previously explained for α -helical coiled-coil domains (Buehler and Ackbarow, 2007; Zhmurov et al., 2012). The transition of α -helical coiled-coil domains into β -sheets under applied force is a well-established phenomenon that was previously described (e.g., Litvinov et al., 2012; Morillas et al., 2001) even in intermediate filament proteins (Fudge et al., 2003; Kreplak et al., 2004). Therefore, it is well established that an intermediate filament stiffens during the application of force. MD simulations explain the force of the structural intermediates along the FE profile that was experimentally detected by AFM. We have revised this in the new version and replaced clamping by sliding (see also answers to reviewer #1) (**pages 6 and 7**).

For the intermediate peak, the average step unit of ≈ 1.9 nm agrees with the structure of dimeric lamin proteins (**Supplementary Fig. 8**). Individual dimers interact to form tetramers of diameter ≈ 3.8 nm (Ahn et al., 2019; Turgay et al., 2017). Thus, we deduced that the intermediate peak arises from dimer-dimer interactions. We have now explained this (see also answers to reviewer #1) in the revised text (**page 7**).

“In the high force regime beyond the yield point (Fig. 1d), lamin filaments showed plastic deformation.”

How is it known that at the yield point it is the lamin filaments, not the nuclear filaments, the nuclear membrane of the nucleoplasmic nuclear basket.

It was shown, proven and re-confirmed in recent studies that lamin filaments at the inner nuclear membrane of *X. laevis* oocyte nucleus form a meshwork (Aebi et al., 1986; Goldberg et al., 2008). No other long filaments are found in these preparations and control experiments are displayed in **Supplementary Figs. 4, 5**. Importantly, only filaments that match the dimensions of lamin filaments as reported before by Turgay *et al.* (Turgay et al., 2017) were observed in *X. laevis* oocyte nuclei (**Supplementary Fig. 8**). The nuclear basket that we have previously studied (Beck et al., 2004) is localized at the NPC and is very short (< 60 nm) compared to the lamin filaments. Also, NPC of the *X. laevis* oocyte show similar dimensions (Eibauer et al., 2015; Frenkiel-Krispin et al., 2010)

We followed the protocols established and used by many labs to open nuclei from *X. laevis* oocyte (mentioned above and **Methods**). For force measurements we selected points on lamin filaments, positioned the AFM cantilever tip on the filaments and pushed on the filaments with forces of up to 8 nN in the closed-loop (feedback on) AFM mode (mentioned in the **Methods – AFM imaging and force spectroscopy**). This protocol provides high-confidence in choosing lamin filaments in the meshwork, while the nuclear pore basket can be easily seen in the images taken before pushing on lamin filaments.

Moreover, pushing on the nuclear membrane or away from the lamin filaments (**Supplementary Figs. 4, 5**) did not show FE curves as observed on lamin filaments. To further rule out the possibility that the filaments were not nucleoskeleton and / or chromatin fibers, we treated opened nuclei with benzonase and imaged the sample. We did not see that the filaments disappeared which would be expected if those were chromatin or filaments of ribonucleoskeleton. The mechanical properties of the filaments did not change in the presence of benzonase (**Supplementary Fig. 14**). Taken together, the results provide solid evidence that we indeed probed lamin filaments and not chromatin or ribo-nucleoskeleton.

Yield point belongs to the FE curve and is not a separate entity. So, if we have shown that the FE curve is from a lamin filament, evidently the yield point is too (as shown for other intermediate filaments). The term yield point is borrowed from materials science and refers to the instance where a lamin filament undergoes plastic or permanent deformation. This is also shown in **Figure 1**. We have now clarified the definition of yield point in the revised text (**page 5**) and also in the legend of **Figure 1**.

Given that 100% of the membrane is not going to be nucleoplasmic side out (these membranes fold back on themselves when the nuclei are opened), maybe it is the outer nuclear membrane or maybe sheets of endoplasmic reticulum that are continuous with the outer nuclear membrane. How is the orientation of the nuclear membrane on the surface determined both globally, and very locally where they are probing?

In this study we only used nuclear envelopes that exhibit a clear view of ‘nucleoplasmic side out’ so that the cantilever tip can ‘see’ lamins as in **Fig. 1a**. This was confirmed by imaging before every lamin pushing experiment.

The reviewer is correct that sometimes the membrane may fold back although we did not see any evidence of that in our study or in the literature. However, this is not a matter of experimental concern in our set-up. It is obvious to determine whether the nucleoplasmic side or the cytoplasmic side is facing up. The most evident proof that the nucleoplasmic side is facing up is the presence of lamin filaments. If the cytoplasmic side of the nuclear membrane is facing the cantilever tip, no lamin filaments would be observed. Moreover, the appearance of NPCs is different from the cytoplasmic and the nucleoplasmic sides. For example, Hoogenboom and co-workers (Stanley et al., 2018) showed recently the distinct views of the nuclear membrane from nucleoplasmic and cytoplasmic sides when they studied the two faces of the NPC.

However, in line with the reviewer’s comment, we have now mentioned this in the **Methods (page 16)**. Since we always imaged the membrane prior to our measurements, we are certain that all our measurements were conducted on the INM. Importantly, if there are folds and ruffles in the membrane, the lamin meshwork will be broken. AFM is sensitive to detect such membrane folds. In the AFM images of the lamin meshwork we took before pushing on the filament, no ruffles were observed. Also, we imaged areas approximately $1\ \mu\text{m} \times 1\ \mu\text{m}$ to clearly see the lamin filaments before applying a force on them. We have explained these points in the revised manuscript.

Page 5: “Interestingly, $\approx 60\%$ of the FE curves also showed a second peak with a step unit of $\approx 4\ \text{nm}$.” But this is a correlation, and a weak one at best. There are many things in the cell at $4\ \text{nm}$ including proteins and the thickness of the membrane.

As discussed above, the lamin filaments were first imaged and then analyzed mechanically; thus we see the structures that were analyzed. Moreover, in these preparations the only observed filaments (by cryo-EM and AFM) of the purified NE are the LIII lamin filaments. Additionally, we performed force-clamp experiments. Force-clamp provides the lifetime of single bonds and also of protein structures. At a constant force, we observed multiple steps with similar life-times. If it was only one step from lamin and other steps from other proteins, steps of similar lifetimes would be unlikely. That more than one protein and also the nuclear membrane have similar lifetimes in the same sample when subjected to constant loads is highly improbable. Our controls (see **Supplementary Fig. 4**) do not show any peaks when the cantilever tip was pushed on the nuclear membrane. This shows that neither the nuclear membrane nor any protein on the membrane showed $4\ \text{nm}$ steps. Taken together, these results provide strong evidence that we characterized lamin filaments and no other proteins or the nuclear membrane. We have referred to **Supplementary Fig. 4** on **page 8** (revised text) where we have mentioned the second peak.

Page 5: “The outer and inner nuclear membranes of frog oocytes are ≈ 50 nm apart, while NPCs are ≈ 90 nm tall structures³⁴ with flexible lamin filaments situated on the cytoplasmic face. This enabled pushing of lamin filaments up to 100 nm towards the glass surface.” I do not understand how the authors reached this conclusion.

In light of the reviewer’s comment, we have added a new figure (**Supplementary Fig. 3**) showing the precise dimension of the NE as measured by cryo-ET and a schematic view of the experimental set up. This figure indicates that we pushed the filaments >90 nm under an applied force. The average distance measured until filament rupture was in fact 91 nm as mentioned in the legend of Figure 2I.

Page 6: “To answer this, we applied a repetitive force protocol to measure the energy dissipated during filament failure. Independent of the loading rate, the energy dissipated in the low force regime was determined to be $\approx 10^{-17}$ J (Fig. 3a, c), increasing to $\approx 10^{-16}$ J ($10^5 k_B T$) up to the apparent failure (Fig. 3b, d). Majority (89%) of the FE curves ($n = 133$) showed reversibility of the plateau up to 0.5 nN indicative of fast refolding (< 50 ms) of the α -helical coiled-coil domain.”

It is not clear to me what the logic was of the experiments or how the authors reached this conclusion.

Dahl *et al.* (J Cell Sci 2004; ref 39) proposed that lamin meshwork is a shock absorber. We wanted to understand if this is an emergent property of the lamin meshwork, *i.e.*, if a single lamin filament is capable of absorbing mechanical shocks or only the meshwork made of many filaments. Further, the experiments by Dahl *et al.* were done on entire intact nuclei that also has contribution from nuclear membrane, chromatin and other nucleoplasmic proteins and factors. We therefore measured the shock absorbing capacity directly on individual lamin filaments without contribution from chromatin or nucleoplasm components. It is mentioned in the manuscript but we have added another sentence in the revised text in light of the reviewer’s comment.

The reason for doing a repetitive protocol was two-fold: (1) to measure the energy dissipated in the low force regime (plateau region) (similar experiments were conducted on vimentin, (Block *et al.*, 2018)). (2) The repetitive protocol also enabled us to answer that the low force regime is reproducible only when the filament is pushed up to 500 pN but not after the filament had been exposed to nanoNewton forces where the structural changes (α -helix to β -sheet transition) is irreversible. This forms the basis for suggesting that there is re-folding of α -helical coiled-coil because we know from simulations that α -helical coiled coils unfold in the low force regime. This would also mean that the filament is not damaged after pushing up to 500 pN and would go back to its original structure to protect the meshwork. We have now added this in the revised text (**page 9**).

In **Supplementary Fig. 10** we showed that the failure force of lamin filaments is independent of loading rate. We therefore determined if the energy dissipated is also independent of loading rate. Indeed, we measured that the energy of dissipation does not increase at higher loading rates as would be expected for a viscous or viscoelastic material. The non-dependence on loading rate suggests that the lamin is an elastic material. This was already mentioned in the text.

We apologize for a typo in the refolding time – it should be 500 ms and not 50 ms. We have corrected this now. The time is calculated based on the speed of relaxation after the first pushing. This is explained in **Methods** (see **Reversible pushing of lamin filaments**).

Page 7: “The similar values of the step sizes as in constant velocity experiments indicate that the 1.3 nm steps signify the failure of single α -helical coiled-coils in a filament, and the 4 nm steps at ≥ 2 nN denote the failure of tetramers.”

This seems like a leap of faith. Why assume it is from the lamins?

We thank the reviewer for raising the point. Based on the argumentations above and the rich literature on the nuclear phase of manually opened nuclear envelopes from *Xenopus* oocytes, it is clear that we are measuring the LIII lamin filaments. It is already well established in the field based on biochemical, SEM, cryo-EM studies that opening *X. laevis* oocyte nuclei predominantly shows lamin meshwork perforated with NPCs.

“The lifetime, τ break, of the α -helical coiled-coils at $F_{load} \leq 1$ nN was measured to be a few hundred milliseconds, suggesting that the coiled-coil structure is the first force buffer. As the force is increased to 3 nN, the α -helix to β -sheet stiffening increases the load bearing capacity of the filament, and the failure requires tens of milliseconds (Supplementary Fig. 10).” How, based on the information shown here, have they assigned the force to the lamins, and then how have they assigned which force is to the alpha helices and which to the beta sheet?

The methodology of choosing the filament for force-clamp was the same as for pushing the filament at a constant velocity. Based on the simulations and forces observed in constant velocity experiments, we proposed that when the force on lamin filaments is held (clamped) at 3 nN, there is α -helix to β -sheet transition. Transition of α -helix to β -sheet is a well-known phenomenon when exerting force on intermediate filaments, even when paracrystalline lamin assemblies were analyzed (Zingerman-Koladko et al., 2016).

The assignment of lamin filaments is already covered in length in the previous and important clarifications asked by the reviewer.

“Interestingly, both, *X. laevis* oocyte and the mammalian nuclear lamin meshworks, exhibit similar topological features of the laminae.” What are the criteria for calling them “similar”. The slopes of c and g in figure 5 look different to me. Is there a statistical analysis they can use? Can they examine other features of the cell to give a sense of what is similar and what is different? I find this sentence and the data, as presented, as uninformative. What if a similar analysis was applied to microtubules or actin under the membrane?

We thank the reviewer for pointing out this point. The text is probably not clear because of technical terms. The topology of the lamin meshworks is similar based on quantitative parameters. The data fit to a Power law that gave the same degree of connectivity for both meshworks. We have explained this in the revised text (**page 11**).

Actin filaments and microtubules are very different in their characteristics. These filaments are not elastic but rather plastic. Actin filaments fail at ~ 100 pN (Kishino and Yanagida, 1988) and microtubules require ~ 500 pN (Schaap et al., 2006). Both are much lower forces than required

for lamins and other intermediate filaments although both actin and microtubule are thicker than lamins.

While lamin meshworks are very similar all around the nucleus, actin networks vary substantially depending on cells and the position within a cell. The actin networks shown below in the figure comprised of actin bundles ~300nm thick, typical for focal adhesion, while at leading edges a branched network was found (Mueller et al., 2017).

Surface rendered view of cell periphery MEF cells. The images show focal adhesion sites. The diameter of each filament is ~8 nm and the network is ~300 nm in thickness.

“and points to the importance of hubs as important signal transmission points in the meshwork.” Where is any hint that these are involved as signaling? This sentence comes completely out of context.

We thank the reviewer for raising this point. We have revised this speculative statement as studying signaling by lamins and binding proteins is in its infancy. Nevertheless, the hubs may be important for the integrity of the meshwork.

References

- 1) Aebi, U., Cohn, J., Buhle, L., and Gerace, L. (1986). The nuclear lamina is a meshwork of intermediate-type filaments. *Nature* *323*, 560-564.
- 2) Ahn, J., Jo, I., Kang, S.M., Hong, S., Kim, S., Jeong, S., Kim, Y.H., Park, B.J., and Ha, N.C. (2019). Structural basis for lamin assembly at the molecular level. *Nat Commun* *10*, 3757.
- 3) Beck, M., Förster, F., Ecke, M., Plitzko, J.M., Melchior, F., Gerisch, G., Baumeister, W., and Medalia, O. (2004). Nuclear pore complex structure and dynamics revealed by cryoelectron tomography. *Science* *306*, 1387-1390.
- 4) Block, J., Witt, H., Candelli, A., Danes, J.C., Peterman, E.J.G., Wuite, G.J.L., Janshoff, A., and Köster, S. (2018). Viscoelastic properties of vimentin originate from nonequilibrium conformational changes. *Sci Adv* *4*: *eaat1161*.
- 5) Buehler, M.J., and Ackbarow, T. (2007). Fracture mechanics of protein materials. *Materials Today* *10*, 46-58.
- 6) Dahl, K.N., Kahn, S.M., Wilson, K.L., and Discher, D.E. (2004). The nuclear envelope lamina network has elasticity and a compressibility limit suggestive of a molecular shock absorber. *J Cell Sci* *117*, 4779-4786.
- 7) Eibauer, M., Pellanda, M., Turgay, Y., Dubrovsky, A., Wild, A., and Medalia, O. (2015). Structure and gating of the nuclear pore complex. *Nat Commun* *6*.
- 8) Frenkiel-Krispin, D., Maco, B., Aebi, U., and Medalia, O. (2010). Structural analysis of a metazoan nuclear pore complex reveals a fused concentric ring architecture. *J Mol Biol* *395*, 578-586.
- 9) Fudge, D.S., Gardner, K.H., Forsyth, V.T., Riekel, C., and Gosline, J.M. (2003). The mechanical properties of hydrated intermediate filaments: insights from hagfish slime threads. *Biophys J* *85*, 2015-2027.
- 10) Goldberg, M.W., Huttenlauch, I., Hutchison, C.J., and Stick, R. (2008). Filaments made from A- and B-type lamins differ in structure and organization. *J Cell Sci* *121*, 215-225.
- 11) Grossman, E., Dahan, I., Stick, R., Goldberg, M.W., Gruenbaum, Y., and Medalia, O. (2012). Filaments assembly of ectopically expressed *Caenorhabditis elegans* lamin within *Xenopus* oocytes. *J Struct Biol* *177*, 113-118.
- 12) Guzmán, C., Jeney, S., Kreplak, L., Kasas, S., Kulik, A.J., Aebi, U., and Forró, L. (2006). Exploring the mechanical properties of single vimentin intermediate filaments by atomic force microscopy. *J Mol Biol* *360*, 623-630.
- 13) Kishino, A., and Yanagida, T. (1988). Force measurements by micromanipulation of a single actin filament by glass needles. *Nature* *334*, 74-76.
- 14) Kreplak, L., Bär, H., Leterrier, J.F., Herrmann, H., and Aebi, U. (2005). Exploring the mechanical behavior of single intermediate filaments. *J Mol Biol* *354*, 569-577.
- 15) Kreplak, L., Doucet, J., Dumas, P., and Briki, F. (2004). New aspects of the alpha-helix to beta-sheet transition in stretched hard alpha-keratin fibers. *Biophys J* *87*, 640-647.
- 16) Kreplak, L., Herrmann, H., and Aebi, U. (2008). Tensile properties of single desmin intermediate filaments. *Biophys J* *94*, 2790-2799.
- 17) Litvinov, R.I., Faizullin, D.A., Zuev, Y.F., and Weisel, J.W. (2012). The α -helix to β -sheet transition in stretched and compressed hydrated fibrin clots. *Biophys J* *103*, 1020-1027.

- 18) Makarov, A.A., Zou, J., Houston, D.R., Spanos, C., Solovyova, A.S., Cardenal-Peralta, C., Rappsilber, J., and Schirmer, E.C. (2019). Lamin A molecular compression and sliding as mechanisms behind nucleoskeleton elasticity. *Nat Commun* 10.
- 19) Moir, R.D., Donaldson, A.D., and Stewart, M. (1991). Expression in *Escherichia coli* of human lamins A and C: influence of head and tail domains on assembly properties and paracrystal formation. *J Cell Sci* 99, 363-372.
- 20) Morillas, M., Vanik, D.L., and Surewicz, W.K. (2001). On the mechanism of alpha-helix to beta-sheet transition in the recombinant prion protein. *Biochemistry* 40, 6982-6987.
- 21) Mueller, J., Szep, G., Nemethova, M., de Vries, I., Lieber, A.D., Winkler, C., Kruse, K., Small, J.V., Schmeiser, C., Keren, K., *et al.* (2017). Load adaptation of lamellipodial actin networks. *Cell* 171, 188-200.
- 22) Qin, Z., and Buehler, M.J. (2011). Flaw tolerance of nuclear intermediate filament lamina under extreme mechanical deformation. *ACS Nano* 5, 3034-3042.
- 23) Rowat, A.C., Foster, L.J., Nielsen, M.M., Weiss, M., and Ipsen, J.H. (2005). Characterization of the elastic properties of the nuclear envelope. *J R Soc Interface* 2 63-69.
- 24) Schaap, I.A.T., Carrasco, C., de Pablo, P.J., MacKintosh, F.C., and Schmidt, C.F. (2006). Elastic response, buckling, and instability of microtubules under radial indentation. *Biophys J* 91, 1521-1531.
- 25) Shimi, T., Kittisopikul, M., Tran, J., Goldman, A.E., Adam, S.A., Zheng, Y., Jaqaman, K., and Goldman, R.D. (2015). Structural organization of nuclear lamins A, C, B1, and B2 revealed by superresolution microscopy. *Mol Biol Cell* 26, 4075-4086.
- 26) Stanley, G.J., Fassati, A., and Hoogenboom, B.W. (2018). Atomic force microscopy reveals structural variability amongst nuclear pore complexes. In *Life Sci Alliance*.
- 27) Stick, R., and Goldberg, M.W. (2010). Oocytes as an experimental system to analyze the ultrastructure of endogenous and ectopically expressed nuclear envelope components by field-emission scanning electron microscopy. *Methods* 51, 170-176.
- 28) Stuurman, N., Heins, S., and Aebi, U. (1998). Nuclear lamins: their structure, assembly, and interactions. *J Struct Biol* 122, 42-66.
- 29) Turgay, Y., Eibauer, M., Goldman, A., Shimi, T., Khayat, M., Harush, K.B., Dubrovsky-Gaupp, A., Sapra, K.T., Goldman, R., and Medalia, O. (2017). The molecular architecture of lamins in somatic cells. *Nature* 543, 261-264.
- 30) Zhmurov, A., Kononova, O., Litvinov, R.O., Dima, R.I., Barsegov, V., and Weisel, J.W. (2012). Mechanical transition from α -helical coiled coils to β -sheets in fibrin(ogen). *J Am Chem Soc* 134, 20396-20402.
- 31) Zingerman-Koladko, I., Khayat, M., Harapin, J., Shoseyov, O., Gruenbaum, Y., Salman, A., Medalia, O., and Ben-Harush, K. (2016). The assembly of *C. elegans* lamins into macroscopic fibers. *J Mech Behav Biomed Mater* 63, 35-43.

Reviewers' comments:

Reviewer #1 (Remarks to the Author):

Medalia and colleagues have adequately addressed most of my comments. The authors have made substantial changes to the manuscript since the first version and asking for more experiments would not be realistic, however the authors should control lamin A/C and B expression using western blot (sup figure 19-20) to allow any conclusion on these data, and an additional condition with depletion of all lamin filaments would constitute a crucial control. In addition, while the legends of the sup figure 20 are clear "(wild-type), only A287 type lamins A/C (Lmn B dko) or only lamin B (Lmn A ko)" the sup 19 chart labels are less clear "lamin B (dko)". The authors should rename the labels or detail the sup figure 19 legends to avoid any confusion.

Reviewer #2 (Remarks to the Author):

The manuscript is vastly improved and I thank the authors for the time and effort to consider all of the comments.

There is one remaining issue that I think would greatly strengthen the paper: If the authors could clearly layout in the manuscript the groups on which they claim that their in vitro measurements have bearing on the behavior of the filaments in the cell. Citing one paper from 2005 to say that the lamins behave the same in vivo and in vitro is not as powerful as clearly stating what are the standards that they used to convince themselves of the utility of the in vitro measurements. There is a long history in the biological literature of people who work on the nucleus who find that the local environment is absolutely essential for understanding the behavior of these molecules. There are a number of reports that there are very different nuclear mechanics inside of cells: The diffusivity is different, the colloid osmotic pressure is different see for example

<https://www.ncbi.nlm.nih.gov/pmc/articles/PMC2797059/>

A critical problem is that with this isolation, the nuclei lose 95% of their proteins in 4 minutes

<https://www.ncbi.nlm.nih.gov/pmc/articles/PMC2112592/>. Add to that the loss of chromatin, there are many reasons to believe that the environment around the isolated lamin is radically different from what happens in the cell. The authors concede: "Ionic strength is known to have an effect on chromatin packing and binding" – why isn't the same true of the lamins? They say that they have tested the effects of nuclease on the behavior of the lamin, but I don't see results there that compel me and that is not the same as comparing the lamin isolated and in situ.

I agree with the authors comment: "On a side note, we would like to mention that historically in vitro analysis of macromolecular assemblies has been very informative and has pushed biology forward." However, for this analysis of each of these in vitro systems, at each step careful studies were done to quantify the behavior in vitro to see if it is recapitulating the behavior in the intact cell. That is why most of the in vitro analysis on assemblies look at structure – it is so difficult and requires care to show that function is conserved.

The current text is still not clear on how do the authors know that they are measuring the outside or inside of the nuclear envelope. They state in the rebuttal: "In this study we only used nuclear

envelopes that exhibit a clear view of 'nucleoplasmic side out' so that the cantilever tip can 'see' lamins as in Fig. 1a. This was confirmed by imaging before every lamin pushing experiment." How was imaging to used to determine orientation. Just saying that they determined it does not help the reader.

There are other statements made by the authors for which the conclusion is clear to them, but they do not make it clear to the reader. For example:

"Based on the argumentations above and the rich literature on the nuclear phase of manually opened nuclear envelopes from *Xenopus* oocytes, it is clear that we are measuring the LIII lamin filaments."

I do not know if the statement is correct or not. I am sure that the authors are convinced. What would make it a stronger manuscript would be if they made clear their reasons to a reader is convinced.

Reviewer #4 (Remarks to the Author):

In the manuscript 'Nonlinear mechanics of lamin filaments and the meshwork topology build an emergent nuclear lamina', Sapra et al. use atomic force microscopy to directly measure the mechanical response of single lamin filaments in their three-dimensional meshwork, by opening up intact nuclei. The authors employ cryo-electron tomography to gain a physical description of the lamina network organization, and apply network analysis and molecular dynamics simulations to explain the complex mechanical behavior of the filaments.

lamins are primarily responsible for the mechanical stability of the nucleus in multicellular organisms and the authors successfully attain measurements of the mechanical properties of individual lamin filaments within the complex scenario of nucleus, a first achievement on its own, seeing as vertebrate lamins cannot be reconstituted into filaments *in vitro*, but rather assemble into less physiologically relevant para-crystalline arrays.

The authors used nuclei from *Xenopus laevis* oocyte for their main experiments, because it is a more feasible model system in terms of sample preparation and the known lack of interaction of this lamina system with chromatin, allowing them to disentangle complex mechanical behavior. However, it remains unclear how representative this systems is, as it have been previously repeatedly shown that the *X. laevis* lamina consists of orthogonally arranged para-crystals. In contrast, the few data published from mammalian cell lines, clearly show more randomly arranged networks. Along these lines, the data provided in the manuscript clearly show an orthogonal arrangement in the AFM images (Figure 1c). In great contrast, the very sparse cryo-ET data provided in the manuscript from the same system shows a more random arrangement of the filaments (surface rendering in Figure 1b and Supplementary Figure 6).

Specifically, in Supplementary Figure 6, the lamina filaments appear somewhat collapsed onto the NPCs. The authors should explain this appearance and how representative it is, as it differs significantly from their own AFM images and previous publications showing an orthogonal pattern. It is recommended that the authors provide a number of datasets (cryo-ET and AFM images) in the

supplementary materials to show the consistency of their observations.

If mechanical measurements have been done on both architectures, this should be clearly introduced into the manuscript, as I suspect that an orthogonal organization of larger filaments bundle may exhibit a different mechanical response of the whole network in comparison to a random network.

The cryo-ET data (Figure 5) also show local bundling of the filaments, as well as different local volume occupancy (concentration). Would bundling fractions or concentration not affect the mechanical behaviour of the network and how likely that in the AFM measurements a single filament is actually probed if bundling exists?

The authors expand their work to mouse embryonic fibroblasts (MEF) that express a single lamin isoform (lamin A or lamin B), supporting that altering lamin type affects mechanics, likely due to changes in the meshwork topology and filaments interactions, rather than the lamin isoform itself. I congratulate the authors on these efforts. However, here they only speculate that a change in network topology occurs to explain the differences in whole nuclei mechanics, but do not support this with structural data from cryo-ET. They additionally do not further attempt to measure the mechanics of single filaments under these conditions (or did I miss this?), which would have been of significant importance to support their claim.

To do experiments on MEF nuclei, the authors employed low salt buffers to swell the cells and nuclei. They continuously claim that these conditions should not affect the lamina. However, it has been reported in the literature that low salt conditions can be used to depolymerize intermediate filaments. The authors further write (line 112) that 'nuclei that were not chemically fixed during deroofing, filamentous meshwork was not observed (Supplementary Figs. 1h, i) and therefore we could not assign filaments and characterize the mechanical properties', which further substantiates the effect of low salt buffer on the lamina. The author should simply state this as it is and should not argue that the lamina remains intact unless they can substantiate this with data.

in addition, their description of the effect of the mutations is somewhat confusing; for example, in line 364 '(ii) nuclei with either lamin A (lamin B knock-out) or lamin B (lamin A knock-out) alone showed higher counter force than wild-type nuclei (Supplementary Fig. 20)', is somewhat contradictory to the statement in line 401 'Point mutations in lamin A transform the nucleus from a resilient to a fragile material'. The authors should strive to make the manuscript more easily understandable to a broader audience than the biomechanics community.

Finally, the complex behavior of the individual filaments measured by the authors is of high interest and has been described for different molecules. I understand that it is an accepted model that the alpha-helices could unfold and transition to beta-sheets that confer higher stiffness. I however wonder if that has ever been experimentally determined by combining AFM measurements, or more microscopic stretching measurements, with spectroscopic measurements of secondary structure transitions? If these interpretations are solely based on the MD simulations, it should be better described in the text.

A few additional comments:

Statistical analysis of (non-)significance should be provided for SFig. 9 and SFig. 14. Especially in 14, it

appears that the mean of the histograms might change following Bensonaze treatment. Authors should validate significance of the change.

In line 175: 'It is suggested that the mechanical reaction of lamin filaments is a robust characteristic and may be key to its function during cell migration through narrow crevices'. It is known that cells that invade through narrow spaces in tissues have much softer nuclei that are a result of differential/reduced lamin expression. The authors should state this more accurately.

The authors continuously refer to Supplementary Fig. 7 as describing lamins in an orthogonal meshwork. I however fail to understand where that organization is implemented into the network; I can only see a random meshwork in the figure.

The authors' main conclusion in line 348, 'These results suggest that the lamin meshwork is an emergent structure; that is, the meshwork is more than the sum of its parts', is of importance and has been demonstrated for a number of hierarchical assemblies in biology. The authors should put their findings in the context of the previous work.

To the editors:

Re: Nonlinear mechanics of lamin filaments and the meshwork topology build an emergent nuclear lamina

We would like to thank the reviewers for their constructive comments and time. We have revised the manuscript according to their comments. Especially, we would like to point out that we added the requested controls (Western blot and fluorescence microscopy imaging that verify the lamin isoform knock-out cells, Supplementary Fig. 19), few sentences on *in vitro* vs. *in situ* behavior of lamins, as well as the effect of low salt on lamin filaments (main text). We have also added additional AFM and cryo-ET images of spread nuclear envelopes (Supplementary Figs. 2, 8). With all these changes and additional modifications that were incorporated as requested (below), we believe the manuscript is improved and hope it can be accepted for publication.

Point by point answers to Reviewers' comments:

Reviewer #1 (Remarks to the Author):

Medalia and colleagues have adequately addressed most of my comments. The authors have made substantial changes to the manuscript since the first version and asking for more experiments would not be realistic, however the authors should control lamin A/C and B expression using western blot (sup figure 19-20) to allow any conclusion on these data, and an additional condition with depletion of all lamin filaments would constitute a crucial control. In addition, while the legends of the sup figure 20 are clear “(wild-type), only A287 type lamins A/C (Lmn B dko) or only lamin B (Lmn A ko)” the sup 19 chart labels are less clear “lamin B (dko)”. The authors should rename the labels or detail the sup figure 19 legends to avoid any confusion.

We thank the reviewer for appreciating the effort and the additional work we have added to the manuscript.

As the reviewer suggested we have added Western blot analysis as well as immunofluorescence images, that indicating the presence and absence of lamin A/C and lamin B in the mammalian cells used (Supp. Fig. 19e).

The labels in Supp. Fig. 19 were modified in light of the reviewer comments.

It is important to note that depletion of all lamin isoforms is possible only in embryonic stem cells and not with MEFs, which we have used for our experiments.

Reviewer #2 (Remarks to the Author):

Reviewer: The manuscript is vastly improved and I thank the authors for the time and effort to consider all of the comments.

We thank the reviewer for appreciating our work and efforts as well as for the constructive comments. We have addressed the remaining concerns of the reviewer.

There is one remaining issue that I think would greatly strengthen the paper: If the authors could clearly layout in the manuscript the groups on which they claim that their *in vitro* measurements have bearing on the behavior of the filaments in the cell. Citing one paper from 2005 to say that the lamins behave the same *in vivo* and *in vitro* is not as powerful as clearly stating what are the standards that they used to convince themselves of the utility of the *in vitro* measurements. There is a long history in the biological literature of people who work on the nucleus who find that the local environment is absolutely essential for understanding the behavior of these molecules. There are a number of reports that there are very different nuclear mechanics inside of cells: The diffusivity is different, the colloid osmotic pressure is different see for example <https://www.ncbi.nlm.nih.gov/pmc/articles/PMC2797059/>

We thank the reviewer for the suggestion. We added additional citations on similar behavior of filaments *in vitro* and *in vivo*. For example, M.F. Carlier and colleagues have shown how actin filament assembly *in vitro* resemble the *in vivo* activity, velocity of the entire motility system (Boujemaa-Paterski et al., 2001; Wiesner et al., 2003). Moreover, lamin meshwork is unaltered when treated with detergents and different salts based on comparison of cryo-ET and 3D-SIM data (Turgay et al., 2017) and the *in situ* measurements by (Mahamid et al., 2016). We also show that the meshwork of lamins in MEFs and isolated NEs are similar (Fig. 5). In another study, the *C. elegans* lamin showed similar structure (~4 nm in diameter) when imaged *in situ* (Harapin et al., 2015) and when expressed in the system used in this work (Grossman et al., 2012) We have mentioned this in Discussion now (**page 12**).

Moreover, IF proteins are known to be very stable and do not change much even when cells are lysed (e.g. our hair and skin retained intermediate filament meshwork in dead cells– keratin – evolved to protect the human body).

We would like to emphasize that our measurements involved isolation of nuclear membrane it is not an *in vitro* approach as understood in the conventional sense of expressing and purifying a protein for measurements. We did not purify nor assemble lamin filaments but measured the ***in situ assembled*** meshwork at its site of origin; hence attached to the nuclear membrane.

Mechanical measurements are commonly done on purified proteins; however we did *in situ* measurements here that has not been attempted at the single molecule level before. Although we do not claim in the manuscript that the lamin filaments have the same mechanical properties inside the cell, it is unlikely that the characteristics change dramatically (e.g., ribosomes synthesize proteins *in vitro* as well as *in vivo*. Actin filament and microtubule characteristics are very similar *in vitro* and *in vivo*).

A critical problem is that with this isolation, the nuclei lose 95% of their proteins in 4 minutes. <https://www.ncbi.nlm.nih.gov/pmc/articles/PMC2112592/>. Add to that the loss of chromatin, there are many reasons to believe that the environment around the isolated lamin is radically different from what happens in the cell. The authors concede: “Ionic strength is known to have an effect on chromatin packing and binding” – why isn’t the same true of the lamins? They say that they have tested the effects of nuclease on the behavior of the lamin, but I don’t see results there that compel me and that is not the same as comparing the lamin isolated and *in situ*.

We thank the reviewer for the interesting reference. We have previously shown that gentle extraction does not change the lamin meshwork as determined by super resolution microscopy (see (Turgay et al., 2017), **Fig. 1C**, (Shimi et al., 2015)). In addition, several previous studies showed that lamin LIII meshwork can be maintained *in situ* following purification (Aebi et al., 1986; Goldberg et al., 2008). We followed similar protocols for the present study.

The reference mentioned by the reviewer clearly states (page 1, 2nd column, 2nd para): “*Most nuclear proteins in vivo exist at least partially as diffusive molecules (13), and conventional aqueous isolation procedures take minutes or hours. Hence, any protein which remains in the nucleus following aqueous isolation is likely to be part of the nuclear matrix or other structural elements, or tightly associated with chromosomes.*” Lamins are a part of the nuclear matrix and remain in the nucleus after isolation. The referred paper relies on cryo-freezing. We have also shown by cryo-electron microscopy that lamin filaments stay at the nuclear membrane and was already stated in the manuscript with references.

The referred paper also clearly states that loss of proteins is mainly because of 2 processes: (i) diffusion within the nucleus, (ii) permeation through the nuclear surface. Lamins are attached to the inner nuclear membrane and are large proteins as they form filaments. Lamin filaments do not diffuse similar to other small soluble proteins through the nuclear pore complex – the predominant mechanism of permeation through the nuclear surface. In summary, there is a line of evidence that lamin filaments (similarly to other intermediate filament proteins) are not affected substantially through removal of soluble nuclear proteins.

I agree with the authors comment: “On a side note, we would like to mention that historically *in vitro* analysis of macromolecular assemblies has been very informative and has pushed biology forward.” However, for this analysis of each of these *in vitro* systems, at each step careful studies were done to quantify the behavior *in vitro* to see if it is recapitulating the behavior in the intact cell. That is why most of the *in vitro* analysis on assemblies look at structure – it is so difficult and requires care to show that function is conserved.

We thank the reviewer for the insight. Structural studies are *in vitro*, as the reviewer correctly points out, and not all of them look at the function inside a cell which is an enormously challenging task. To our knowledge, the structure of a protein elucidated *in vitro* has rarely been shown to have the same structure *in vivo*. Most functional assays rely on *in vitro* biochemistry. The entire field of single molecule studies is a shining example of this. Lamin filaments are known to have a mechanical function that we confirm in the present study. We also clearly state in our manuscript that NEs were extracted! We believe the reviewer’s concern is taken care of.

As we have already stated in our previous answers, the goal of the present study was to study the mechanical properties of lamins without any other influence, *i.e.*, chromatin, proteins, and not to compare our measurements with *in vivo*. Our approach is the best strategy for doing this, it is *in situ* and at the same time is not influenced by other proteins as the reviewer would agree that most proteins and chromatin are washed away within minutes (above reference from the reviewer – Paine *et al.*, J Cell Biology 1983). This is the first time that the mechanical properties of lamin filaments were analyzed using *in situ* assembled filament meshwork. Based on our mechanical measurements we have put forth a tentative model of *in vivo* mechanics (**Fig 7**).

The current text is still not clear on how do the authors know that they are measuring the outside or inside of the nuclear envelope. They state in the rebuttal: “In this study we only used nuclear envelopes that exhibit a clear view of ‘nucleoplasmic side out’ so that the cantilever tip can ‘see’ lamins as in Fig. 1a. This was confirmed by imaging before every lamin pushing experiment.” How was imaging to used to determine orientation. Just saying that they determined it does not help the reader.

Fig. 1a is a schematic but Fig. 1b shows a representative image. We have now added this in the caption to Fig 1. We have now added the reference to Fig 1b and added an extra reference 90 (Stanley *et al.*, Life Sci Alliance 2018) on page 16 (**Methods**).

There are other statements made by the authors for which the conclusion is clear to them, but they do not make it clear to the reader. For example:

“Based on the argumentations above and the rich literature on the nuclear phase of manually opened nuclear envelopes from *Xenopus* oocytes, it is clear that we are measuring the LIII lamin filaments.”

I do not know if the statement is correct or not. I am sure that the authors are convinced. What would make it a stronger manuscript would be if they made clear their reasons to a reader is convinced.

The statement is indeed correct. We had already made the reasons clear in the manuscript and cited the references. However, we added additional references that prove (again, (Aebi *et al.*, 1986) showed it for the first time) that lamin LIII are filaments in *Xenopus* in NE preparations (Goldberg *et al.*, 2008).

We have attached below a figure from (Stanley *et al.*, 2018) showing the nucleoplasmic side and cytoplasmic side with and without lamins, respectively.

High-resolution AFM imaging of intact *X. laevis* oocyte NEs in solution. (A) AFM topography of the cytoplasmic side of the NE. White asterisks denote two (out of several) possible appearances of cargo molecules stuck in transit (see the High-resolution AFM imaging of the NE section). The white arrows show instances of NPCs connecting to one another—likely by their cytoplasmic filaments. (B–G) Magnified views of NPCs highlighting the observed variability in the pore lumens. (H) Nucleoplasmic side of the NE. The lamina meshwork is observed as tightly bunched filaments running in tandem around the NPCs, with little or no spacing between them (white arrows show patches of exposed lamin protofilaments). In addition, there are longer filaments (presumably actin, see Fig S4) that interweave around the NPCs, sometimes branching. Inset: apparent branching and termination—and possibly anchoring—of such filaments on the NE. (I) As (H), but with the lamina meshwork appearing more stretched. (J–L) Higher magnification images of NPCs, revealing spoked structures consistent with the nuclear basket. The NPC in (L) is unusually large with a scaffold diameter of 100 ± 4 nm: larger than the usual measured diameter of 85 ± 4 nm ($n = 282$ for nucleoplasmic NPCs; see also Fig S1). Scale bars: 300 nm (A, H, I); 100 nm (B–G; H, inset; and J–L). Colour scales (height, see top right in A): 100 nm (A, H, I), 70 nm (H, inset), 60 nm (B–G), and 65 nm (J–L).

Reviewer #4 (Remarks to the Author):

Reviewer: In the manuscript ‘Nonlinear mechanics of lamin filaments and the meshwork topology build an emergent nuclear lamina’, Sapra et al. use atomic force microscopy to directly measure the mechanical response of single lamin filaments in their three-dimensional meshwork, by opening up intact nuclei. The authors employ cryo-electron tomography to gain a physical description of the lamina network organization, and apply network analysis and molecular dynamics simulations to explain the complex mechanical behavior of the filaments.

Lamins are primarily responsible for the mechanical stability of the nucleus in multicellular organisms and the authors successfully attain measurements of the mechanical properties of individual lamin filaments within the complex scenario of nucleus, a first achievement on its own, seeing as vertebrate lamins cannot be reconstituted into filaments in vitro, but rather assemble into less physiologically relevant para-crystalline arrays.

We thank the reviewer for appreciating the importance of the work.

The authors used nuclei from *Xenopus laevis* oocyte for their main experiments, because it is a more feasible model system in terms of sample preparation and the known lack of interaction of this lamina system with chromatin, allowing them to disentangle complex mechanical behavior. However, it remains unclear how representative this system is, as it has been previously repeatedly shown that the *X. laevis* lamina consists of orthogonally arranged para-crystals. In contrast, the few data published from mammalian cell lines, clearly show more randomly arranged networks. Along these lines, the data provided in the manuscript clearly show an orthogonal arrangement in the AFM images (Figure 1c). In great contrast, the very sparse cryo-ET data provided in the manuscript from the same system shows a more random arrangement of the filaments (surface rendering in Figure 1b and Supplementary Figure 6). Specifically, in Supplementary Figure 6, the lamina filaments appear somewhat collapsed onto the NPCs. The authors should explain this appearance and how representative it is, as it differs significantly from their own AFM images and previous publications showing an orthogonal pattern. It is recommended that the authors provide a number of datasets (cryo-ET and AFM images) in the supplementary materials to show the consistency of their observations. If mechanical measurements have been done on both architectures, this should be clearly introduced into the manuscript, as I suspect that an orthogonal organization of larger filament bundles may exhibit a different mechanical response of the whole network in comparison to a random network.

We believe the reviewer is referring to **Supplementary Figure 8 (cryo-ET data)** and not **Supplementary Figure 6 (simulations)**. We have shown a random lamin meshwork in Supplementary Figure 2c. This is similar to Supplementary Figure 8a which is a slice of the many images obtained during tomography. This is already mentioned in the figure caption. We have now mentioned in the revised manuscript that the measurements were done on both orthogonal and random meshwork (**captions of Figure 1, Supplementary Figure 2**).

The apparent ‘collapse’ of lamin filaments on the NPC seems because the image is a 10 nm slice of the many acquired. The position of lamin is at the level of the NPC. We have added more images – AFM (**Supplementary Figure 2**) and cryo-ET slices (**Supplementary Figure 8**).

The cryo-ET data (Figure 5) also show local bundling of the filaments, as well as different local volume occupancy (concentration). Would bundling fractions or concentration not affect the mechanical behaviour of the network and how likely that in the AFM measurements a single filament is actually probed if bundling exists?

Meshwork analysis on a number of tomograms showed similar network properties which are already reported in the manuscript. There is no bundling because of sample preparation as the lamin meshwork in that case would be contracted. We also do not observe bundling of filaments *per se*. However, there may be 2 filaments adjacent to each other. Our analysis of single filaments takes this into account and elaborate analysis based on the dimensions of the filaments (4 nm) and step size from AFM constant velocity and force-clamp measurements (4 nm) suggest that we measured single lamin filaments. This is already discussed in the manuscript.

We also observed more than 1 peak that could be because of more than one filament (**Supplementary Figure 9, Supplementary Figure 12**). However, the probability of those peaks is low (**Supplementary Figure 13, Supplementary Table 3**). We have mentioned the term ‘bundling’ in the revised text (**page 8**).

The authors expand their work to mouse embryonic fibroblasts (MEF) that express a single lamin isoform (lamin A or lamin B), supporting that altering lamin type affects mechanics, likely due to changes in the meshwork topology and filaments interactions, rather than the lamin isoform itself. I congratulate the authors on these efforts. However, here they only speculate that a change in network topology occurs to explain the differences in whole nuclei mechanics, but do not support this with structural data from cryo-ET. They additionally do not further attempt to measure the mechanics of single filaments under these conditions (or did I miss this?), which would have been of significant importance to support their claim.

We agree with the reviewer that cryo-ET experiments to show changes in meshwork topology in MEF nuclei is important and is a part of our ongoing efforts. However, that was not the focus of the current work and is an elaborate study on its own. Based on recent works by others (Funkhouser et al., 2013; Shimi et al., 2015) we suggested that nuclear mechanics in our experiments changed because of perturbation in lamin meshwork. However, as pointed by the reviewer, we have now added that lamin isoform interactions with nuclear proteins could be a reason for the observed change in the mechanical response and cited Shimi et al., 2015 (**page 12**).

The reviewer indeed missed in the manuscript our attempt to measure single filaments in MEF nuclei. To demarcate a single filament in unfixed MEF nuclei is difficult and therefore we did not continue with that approach. This is already mentioned in the manuscript (**page 4**).

To do experiments on MEF nuclei, the authors employed low salt buffers to swell the cells and nuclei. They continuously claim that these conditions should not affect the lamina. However, it has been reported in the literature that low salt conditions can be used to depolymerize intermediate

filaments. The authors further write (line 112) that ‘nuclei that were not chemically fixed during deroofing, filamentous meshwork was not observed (Supplementary Figs. 1h, i) and therefore we could not assign filaments and characterize the mechanical properties’, which further substantiates the effect of low salt buffer on the lamina. The author should simply state this as it is and should not argue that the lamina remains intact unless they can substantiate this with data.

Our claim is substantiated with data as follows: We performed the de-roofing experiments in 2 ways: (1) fixed the nuclei **after de-roofing**, (2) did not fix the nuclei after de-roofing. The swelling step in both cases was in a low-salt buffer. However, in the fixed nuclei we observed clear filaments whereas in the unfixed ones we did not (mentioned in the manuscript and **Supplementary Figure 1**) (also the reason we did not do single filament experiments on unfixed nuclei – answered above). Therefore, we swelling step in the low salt buffer did not depolymerize the lamin filaments. Moreover, the *C. elegans* lamin showed similar 4 nm thick filamentous structure when image in situ (Harapin et al 2015) and when expressed in *Xenopus* oocytes (using NE extraction in low salt). This is another evident that lamins are stable in low salt buffer.

Further, lamins can only disassemble by phosphorylation and are insensitive to ionic strength and cannot disassemble (Beaudouin et al., 2002) (Zwenger et al., 2015).

In addition, their description of the effect of the mutations is somewhat confusing; for example, in line 364 ‘(ii) nuclei with either lamin A (lamin B knock-out) or lamin B (lamin A knock-out) alone showed higher counter force than wild-type nuclei (Supplementary Fig. 20)’, is somewhat contradictory to the statement in line 401 ‘Point mutations in lamin A transform the nucleus from a resilient to a fragile material’. The authors should strive to make the manuscript more easily understandable to a broader audience than the biomechanics community.

We have now removed the sentence ‘Point mutations...fragile material’ (**page 13**). We thank the reviewer for the suggestion.

Finally, the complex behavior of the individual filaments measured by the authors is of high interest and has been described for different molecules. I understand that it is an accepted model that the alpha-helices could unfold and transition to beta-sheets that confer higher stiffness. I however wonder if that has ever been experimentally determined by combining AFM measurements, or more microscopic stretching measurements, with spectroscopic measurements of secondary structure transitions? If these interpretations are solely based on the MD simulations, it should be better described in the text.

There is a body of knowledge showing α -helix to β -sheet transition in coiled-coil proteins. It is a combination of AFM, spectroscopic experiments and MD simulations as the reviewer mentioned. We have emphasized the point and added relevant citations in the revised manuscript (**page 8**).

A few additional comments:

Statistical analysis of (non-)significance should be provided for SFig. 9 and SFig. 14. Especially in 14, it appears that the mean of the histograms might change following Benson treatment. Authors should validate significance of the change.

Because the standard deviations are large and overlapping, we do not think it is sound to mention the statistical significance. We have added the means of the histograms with and without benzonase treatment (**Supplementary Figure 14**). It should be noted that the histograms are overlapping and a few outliers could easily skew the mean.

In line 175: ‘It is suggested that the mechanical reaction of lamin filaments is a robust characteristic and may be key to its function during cell migration through narrow crevices’. It is known that cells that invade through narrow spaces in tissues have much softer nuclei that are a result of differential/reduced lamin expression. The authors should state this more accurately.

We thank the reviewer for the suggestion and have revised this sentence in the manuscript (**page 6**). However, in the context of the sentence based on the simulations with different probe radius (**Supplementary Figure 6**), to mention softer nuclei as a result of differential lamin expression is not meaningful.

The authors continuously refer to Supplementary Fig. 7 as describing lamins in an orthogonal meshwork. I however fail to understand where that organization is implemented into the network; I can only see a random meshwork in the figure.

We indeed mentioned it once on **page 7** and have also cited the reference – Qin et al., *ACS Nano* 2011. The representation is indeed misleading and we have modified it. The authors’ main conclusion in line 348, ‘These results suggest that the lamin meshwork is an emergent structure; that is, the meshwork is more than the sum of its parts’, is of importance and has been demonstrated for a number of hierarchical assemblies in biology. The authors should put their findings in the context of the previous work.

We thank the reviewer for suggesting this. We have now added an extra sentence with references in the revised manuscript (**page 12**).

References

- Aebi, U., Cohn, J., Buhle, L., and Gerace, L. (1986). The nuclear lamina is a meshwork of intermediate-type filaments. *Nature* *323*, 560-564.
- Beaudouin, J., Gerlich, D., Daigle, N., Eils, R., and Ellenberg, J. (2002). Nuclear envelope breakdown proceeds by microtubule-induced tearing of the lamina. *Cell* *108*, 83-96.
- Boujemaa-Paterski, R., Guoin, E., Hansen, G., Samarin, S., Le Clainche, C., Didry, D., Dehoux, P., Cossart, P., Kocks, C., Carlier, M.F., *et al.* (2001). Listeria protein ActA mimics WASp family proteins: it activates filament barbed end branching by Arp2/3 complex. *Biochemistry* *40*, 11390-11404.
- Funkhouser, C.M., Sknepnek, R., Shimi, T., Goldman, A.E., Goldman, R.D., and Olvera de la Cruz, M. (2013). Mechanical model of blebbing in nuclear lamin meshworks. *Proc Natl Acad Sci USA* *110*, 3248-3253.
- Goldberg, M.W., Huttenlauch, I., Hutchison, C.J., and Stick, R. (2008). Filaments made from A- and B-type lamins differ in structure and organization. *J Cell Sci* *121*, 215-225.
- Grossman, E., Dahan, I., Stick, R., Goldberg, M.W., Gruenbaum, Y., and Medalia, O. (2012). Filaments assembly of ectopically expressed *Caenorhabditis elegans* lamin within *Xenopus* oocytes. *Journal of structural biology* *177*, 113-118.
- Harapin, J., Bormel, M., Sapra, K.T., Brunner, D., Kaech, A., and Medalia, O. (2015). Structural analysis of multicellular organisms with cryo-electron tomography. *Nature methods* *12*, 634-636.
- Mahamid, J., Pfeffer, S., Schaffer, M., Villa, E., Danev, R., Cuellar, L.K., Forster, F., Hyman, A.A., Plitzko, J.M., and Baumeister, W. (2016). Visualizing the molecular sociology at the HeLa cell nuclear periphery. *Science* *351*, 969-972.
- Shimi, T., Kittisopikul, M., Tran, J., Goldman, A.E., Adam, S.A., Zheng, Y., Jaqaman, K., and Goldman, R.D. (2015). Structural organization of nuclear lamins A, C, B1, and B2 revealed by superresolution microscopy. *Mol Biol Cell* *26*, 4075-4086.
- Stanley, G.J., Fassati, A., and Hoogenboom, B.W. (2018). Atomic force microscopy reveals structural variability amongst nuclear pore complexes. In *Life Sci Alliance*.
- Turgay, Y., Eibauer, M., Goldman, A., Shimi, T., Khayat, M., Harush, K.B., Dubrovsky-Gaupp, A., Sapra, K.T., Goldman, R., and Medalia, O. (2017). The molecular architecture of lamins in somatic cells. *Nature* *543*, 261-264.
- Wiesner, S., Helfer, E., Didry, D., Ducouret, G., Lafuma, F., Carlier, M.F., and Pantaloni, D. (2003). A biomimetic motility assay provides insight into the mechanism of actin-based motility. *J Cell Biol* *160*, 387-398.
- Zwinger, M., Roschitzki-Voser, H., Zbinden, R., Denais, C., Herrmann, H., Lammerding, J., Grutter, M.G., and Medalia, O. (2015). Altering lamina assembly reveals lamina-dependent and -independent functions for A-type lamins. *Journal of cell science* *128*, 3607-3620.

Reviewers' comments:

Reviewer #1 (Remarks to the Author):

The authors have adequately addressed all my previous comments. It has come to my attention that the legend from figure 7 may be confusing and it would be better if the authors specify more clearly that the panel a (cryo-ET) was published in a previous paper. Maybe they should say "adapted from ref. 4" (instead of only quoting the reference).

Reviewer #2 (Remarks to the Author):

One of my main concerns is what I consider still to be the lack of caution in extrapolating the results from in vitro to in vivo. Doing these in vitro experiments are important and valuable. However, it is also valuable for the authors to carefully draw lines on what is in vitro.

The authors cite: "Carlier and colleagues have shown how actin filament assembly in vitro resemble the in vivo activity," Yes, there are some ways in which they are similar, and there are also many ways in which they are not. Either way, those are actin filaments and not intermediate filaments and not the filaments of the FG-Nups nor the filaments of the nuclear lamin.

The authors state: "Lamins are a part of the nuclear matrix and remain in the nucleus after isolation." Yes, there are some there. But have the authors done a quantification to show the percentage that are there after extraction? What about other proteins that were associated with them in the cell? The authors concede that a large percentage of the proteins in the nucleus do indeed dissociate: Losing the proteins bound to actin or microtubules alter their behavior. Why not the same for the lamin filaments – many of which have been shown to be bound to nuclear components.

The authors state: "Lamins are attached to the inner nuclear membrane and are large proteins as they form filaments. Lamin filaments do not diffuse similar to other small soluble proteins through the nuclear pore complex." However, it is known that many proteins of the nuclear pore dissociate from the nucleus. These proteins are large (e.g. Nup98) and they are attached, and yet they dissociate.

The authors state: "In summary, there is a line of evidence that lamin filaments (similarly to other intermediate filament proteins) are not affected substantially through removal of soluble nuclear proteins." The authors have left this reader unconvinced and concerned about the readers strong adherence without stronger experimental evidence.

The authors state: "Our approach is the best strategy for doing this, it is in situ and at the same time is not influenced by other proteins as the reviewer would agree that most proteins and chromatin are washed away within minutes (above reference from the reviewer – Paine et al., J Cell Biology 1983)." I am surprised that a cell whose plasma membrane and cytosol have been stripped away, the nuclear membrane is inverted, is being referred to as "in situ." This seems inappropriate.

IN response to the following point:

“I do not know if the statement is correct or not. I am sure that the authors are convinced. What would make it a stronger manuscript would be if they made clear their reasons to a reader is convinced.

The statement is indeed correct. We had already made the reasons clear in the manuscript and cited the references. However, we added additional references that prove (again, (Aebi et al., 1986) showed it for the first time) that lamin LIII are filaments in *Xenopus* in NE preparations (Goldberg et al., 2008).”

I was very disappointed by their answer. Just to say “it is correct” and “we had already made the reasons clear” after I had stated they were not clear to me indicates that the authors are unable to clarify their comments. Simply stating it is so does not prove it. Simply saying, “we have added extra references” without giving the rationale has, in my mind, seriously weakened their argument.

Reviewer #4 (Remarks to the Author):

The authors have addresses most of my concerns. A few issues remain, that I detail below:

In their rebuttal, the authors insist that low salt treatment does not affect the native lamina assembly state in MEF cells. However, they clearly state both in the rebuttal and the main text, that without fixation prior to the low salt buffer swelling, they cannot detect the filaments. The argument that this does not happen for the *C. elegans* lamins is not enough to explain the case for MEFs. I recommend that the author stay true to the data presented in this manuscript and should not stress that the procedure does not affect the lamina in MEFs. Therefore, the sentence in the discussion: ‘Moreover, the structure of lamin filaments studied in low salt buffer was found to be identical to the filaments visualized in situ’, is incorrect considering the presented data.

The authors state now that they have done measurements on both orthogonally assembled and randomly organized lamina meshwork. Yet, it is still interesting that they do not observe differences in the mechanical behaviour of the network. Orthogonally organized meshwork indicates a level of self-assembly and intermolecular interactions beyond that of a tetramer that I expect to contribute to the mechanical behaviour of the meshwork. I find this point weakly addressed by the authors.

Currently, they only state that measurements were done on both architectures in the legend of figure 1. In the main text, the authors state ‘lamin filaments were arranged in a meshwork exhibiting a rectangular pattern or a less organized architecture interspersed and interacting with the NPCs’ (page 5) and no description of this is provided in the network topology analysis. I understand the authors do not see a difference between the oocyte and the MEF architectures, but I would strongly recommend that the authors make it clear that both architectures also resulted in similar topologies (if that is the case), although I would have liked to see an analysis of the different architectures done independently and then a quantitative comparison between them.

In the section ‘Mechanics of mammalian nuclei’, the authors start by stating that the concentration

of lamins would alter the mechanical properties of the NE. However, it is far from being clear from their experiments of the knockouts that the overall concentrations are per se altered. In addition, the knockouts do not necessarily make the nuclei softer as would be hypothesised based on the contribution of concentrations only. I believe that authors should rephrase the paragraph to focus on the potential contribution of each of the lamin types to a specific network topography (as they now show A-type lamin knockout MEF cells present orthogonal arrangement, while the B-type lamin knockout show random arrangement (Supplementary Fig. 1)).

Statistical non-significance and the confidence values should still be provided despite of the large standard deviations. That is the whole point of using statistical tests to provide a confidence level in the claimed significance or insignificance. I would urge the authors to include a similar analysis in their Supplementary Fig. 20 to support their claims.

Minor comments:

Sup. Note 1: first paragraph, equation is incomplete: [$\theta = \tan^{-1} (d_{low} / (0.5 l))$].

In Supplementary Fig. 11. Authors should clearly indicate what the color bar represents. presumably normalized densities of the populations?

To the editors:

Re: Nonlinear mechanics of lamin filaments and the meshwork topology build an emergent nuclear lamina

We would like to thank you and the reviewers again for the constructive comments and time. We have revised the manuscript according to their comments.

Point by point answers to Reviewers' comments:

Reviewer #1 (Remarks to the Author):

The authors have adequately addressed all my previous comments. It has come to my attention that the legend from figure 7 may be confusing and it would be better if the authors specify more clearly that the panel a (cryo-ET) was published in a previous paper. Maybe they should say "adapted from ref. 4" (instead of only quoting the reference).

We thank the reviewer and are delighted that the reviewer is satisfied. We have made the correction recommended by the reviewer (**page 35**).

Reviewer #2 (Remarks to the Author):

One of my main concerns is what I consider still to be the lack of caution in extrapolating the results from *in vitro* to *in vivo*. Doing these *in vitro* experiments are important and valuable. However, it is also valuable for the authors to carefully draw lines on what is *in vitro*.

We thank the reviewer for recognizing the importance of these experiments. We also appreciate the concern on the extrapolation of our results to *in vivo*. Therefore, we have toned down the extrapolation from *in vitro* to *in vivo*. First, we discuss that our system is not entirely *in situ* as we are removing the nucleus from the cell and this changes the nuclear environment and may alter the mechanical properties (**page 13**). We have now minimized the use of *in situ* and also changed *in situ* to *in vitro* depending on the context. We use the term ‘*in situ* assembled lamin’ at a few places which is correct, and we hope should not be a matter of any further concern.

We have also mentioned in the **Discussion** that the proposed model (**Figure 7b**) is based on *in vitro* experiments and its relevance inside a cell will need further investigation (**page 14, caption to Figure 7**).

The authors cite: “Carlier and colleagues have shown how actin filament assembly *in vitro* resemble the *in vivo* activity,” Yes, there are some ways in which they are similar, and there are also many ways in which they are not. Either way, those are actin filaments and not intermediate filaments and not the filaments of the FG-Nups nor the filaments of the nuclear lamin.

We thank the reviewer for stressing this point. We believe the reviewer is correct that a direct comparison between actin and intermediate filaments may not be relevant here. The statement was a scholastic argument emphasizing the importance of *in vitro* studies that have increased our understanding of the *in vivo* mechanisms. We have removed this reference and the sentence related to actin from the Discussion (**page 13**).

Vertebrate/mammalian lamin filaments cannot be assembled *in vitro* and we still do not know their structure in high resolution. Therefore, any information on structure and properties of lamin filaments is a stepping stone towards understanding these protein assemblies.

The authors state: “Lamins are a part of the nuclear matrix and remain in the nucleus after isolation.” Yes, there are some there. But have the authors done a quantification to show the percentage that are there after extraction? What about other proteins that were associated with them in the cell? The authors concede that a large percentage of the proteins in the nucleus do indeed dissociate: Losing the proteins bound to actin or microtubules alter their behavior. Why not the same for the lamin filaments – many of which have been shown to be bound to nuclear components.

We appreciate the point raised by the reviewer. The sentence quoted by the reviewer is not in the manuscript but in our previous answer to the reviewer’s comments. The reviewer is correct that

not all proteins are associated with lamins, and that removal of associated proteins might change the mechanical properties of the meshwork. We had mentioned this in the manuscript (**page 5, end of 1st para**) and now another sentence is added in **Discussion (page 13)** stating that removing associated proteins may alter the mechanical properties of the meshwork. However, the basic properties of individual lamin filaments does not depend on the percentage of lost/retained lamins (see below).

A number of published papers have shown that >90% of the B-type lamins (LIII is a B-type lamin) are not removed by low or high salt treatments. For example, B-type lamins are known to be insoluble in both hypotonic and hypertonic solutions (Markiewicz et al., 2005; Scott and O'Hare, 2001); lamin LIII from *X. laevis* oocyte nucleus is a B-type lamin.

Figure (Western blot) from Markiewicz et al., 2005 showing that lamins B1 and B2 are resistant to low and high salt treatment

Fig. 3. Solubility properties of lamins and lamina-associated proteins during myogenesis. C2C12 myoblasts were induced to differentiate with 2% horse serum. At 0, 72 and 120 hours after transfer to differentiation medium, cells were harvested and subjected to nuclear isolation. Nuclei were either solubilized in SDS (nuclei) or sequentially extracted with hypotonic (LS) or hypertonic (HS) buffer. Samples were resolved by SDS-PAGE along with material resistant to extraction (INS), transferred to nitrocellulose and **blotted with antibodies** against lamin B1 (a), lamin B2 (b), lamin A (c) and lamin C (d), and the intensity of each band evaluated by densitometry and expressed as a proportion of each protein in whole nuclear extracts.

Figure from Scott and O'Hare, 2001

FIG. 6. (a) Total cell lysates of COS-1 cells infected with HSV-1 at 10 PFU/cell or of control uninfected cells, stained with Coomassie brilliant blue (CBB) or blotted for lamins A/C or B2. Levels of lamin B2 and, more strikingly, lamins A/C are reduced in infected-cell lysates (Inf) compared to uninfected-cell lysates (M). (b) Soluble and insoluble fractions of COS-1 cells infected or uninfected as for panel a blotted for lamins A/C or B2. Lamins are insoluble (insol.) in a low-salt buffer (lanes 1 to 3). As in panel a, there is a loss of lamins in infected COS-1 cells which is most pronounced for lamins A/C. In high-salt buffers, lamins are partially soluble (sol.) (lanes 4 to 6). Loading double quantities (Inf*) reveals that the only detectable lamin A/C in infected cells is in the soluble fraction, while lamin B2 retains the predominantly insoluble profile seen in uninfected cells.

While lamins are unlikely to be removed by the buffer conditions used, we did not claim that associated proteins were not removed. Therefore, we mentioned that associated proteins may detach from the lamin filaments (page 5 end of 1st para, page 13).

We would also like to reiterate the main point, which probably was not clear before, that the lamin meshwork analysis provides a physical model that can be used to understand the mechanical properties of lamin filaments in a meshwork. Further, MD simulations show that a change in the meshwork topology leads to a change in the mechanical properties of the filaments. We used data of a real lamin meshwork as seen by cryo-ET (and also used for AFM measurements) to generate a physical model for MD simulations. The topology of the model was changed to understand the role in the mechanical properties of lamin filaments. These results are novel and further our understanding of lamin mechanics. Even if the meshwork was perturbed because of nucleus isolation and opening, the starting

meshwork was the same in AFM experiments and the one used to create the model for MD simulations. We have mentioned this in the Discussion of the revised manuscript (page 13).

The authors state: “Lamins are attached to the inner nuclear membrane and are large proteins as they form filaments. Lamin filaments do not diffuse similar to other small soluble proteins through the nuclear pore complex.” However, it is known that many proteins of the nuclear pore dissociate from the nucleus. These proteins are large (e.g. Nup98) and they are attached, and yet they dissociate.

The reviewer is correct for NPC proteins. However, the diffusion rate of B-type lamins is of several hours (~2.5 h for lamin B1, (Moir et al., 2000)). Our entire procedure of nucleus isolation, swelling and opening the nuclear membrane takes ~10 mins. In any case, as mentioned above we have now addressed the concern of removal of lamins and associated proteins.

The authors state: “In summary, there is a line of evidence that lamin filaments (similarly to other intermediate filament proteins) are not affected substantially through removal of soluble nuclear proteins.” The authors have left this reader unconvinced and concerned about the readers strong adherence without stronger experimental evidence.

The reviewer is correct that nuclear environment would change the properties of the lamina. For example, chromatin interacts with lamins and together they change the mechanical properties of the entire nucleus (Liu et al., 2014; Stephens et al., 2017). We have focused to study the lamin filament itself, and we mention it in the text. We mentioned that associated proteins may detach from the lamin filaments (**pages 5 and 13**).

We agree with the reviewer that a change in the meshwork topology leads to a change in the overall mechanical properties of the lamina. We have mentioned this in the **Discussion** of the revised manuscript (**page 13**).

The authors state: “Our approach is the best strategy for doing this, it is *in situ* and at the same time is not influenced by other proteins as the reviewer would agree that most proteins and chromatin are washed away within minutes (above reference from the reviewer – Paine et al., J Cell Biology 1983).” I am surprised that a cell whose plasma membrane and cytosol have been stripped away, the nuclear membrane is inverted, is being referred to as “*in situ*.” This seems inappropriate.

We appreciate the reviewer’s concern. As mentioned above we have changed *in situ* to *in vitro* or ‘*in situ* assembled lamin’ in the manuscript.

IN response to the following point:

“I do not know if the statement is correct or not. I am sure that the authors are convinced. What would make it a stronger manuscript would be if they made clear their reasons to a reader is

convinced.

The statement is indeed correct. We had already made the reasons clear in the manuscript and cited the references. However, we added additional references that prove (again, (Aebi et al., 1986) showed it for the first time) that lamin LIII are filaments in *Xenopus* in NE preparations (Goldberg et al., 2008).”

I was very disappointed by their answer. Just to say “it is correct” and “we had already made the reasons clear” after I had stated they were not clear to me indicates that the authors are unable to clarify their comments. Simply stating it is so does not prove it. Simply saying, “we have added extra references” without giving the rationale has, in my mind, seriously weakened their argument.

We apologize for the lack on our part for not making it clear. We have now given explicit reasons in on why we think our statement is correct.

Our procedure involved using a glass microneedle to open the nuclear membrane such that the nucleoplasmic side was facing upward, *i.e.*, inner nuclear membrane (INM). The nuclear contents including chromatin were gently removed, and the stuck nuclear membrane washed with an ample volume of buffer. If the outer nuclear membrane (ONM), *i.e.*, the cytoplasmic side, were facing upward, we would not expect to see lamin filaments but only NPCs when imaged by AFM (Stanley et al., 2018). We have mentioned this in the **Methods (page 16)**.

Reviewer #4 (Remarks to the Author):

The authors have addresses most of my concerns. A few issues remain, that I detail below:

In their rebuttal, the authors insist that low salt treatment does not affect the native lamina assembly state in MEF cells. However, they clearly state both in the rebuttal and the main text, that without fixation prior to the low salt buffer swelling, they cannot detect the filaments.

We apologize for not making the protocol steps clear. The nuclei were swollen in a low salt buffer **before de-roofing, and fixation was done after de-roofing**. We performed the de-roofing experiments in 2 ways: (1) fixed the nuclei **after de-roofing**, (2) did not fix the nuclei after de-roofing. The swelling step in both cases was in hypotonic Ringer's solution. **In brief, the steps were: swelling in hypotonic Ringer's solution → de-roofing → fixation / no fixation**. The procedure is explained in **Methods**.

In the fixed nuclei we observed clear filaments whereas in the unfixed ones we did not (mentioned in the manuscript and **Supplementary Fig. 1**) – the reason we did not do single filament experiments on unfixed nuclei. Therefore, we believe that the swelling step in the low salt buffer did not depolymerize the lamin filaments as the filaments were observed after fixation. Without fixation there could be areas of lipid membranes covering / diffusing on the surface or sticking to the AFM cantilever tip thereby hindering clear imaging of the filaments. Moreover, a number of published papers have shown that >90% of the B-type lamins (LIII is a B-type lamin) are not removed by low or high salt treatments. For example, B-type lamins are known to be insoluble in both hypotonic and hypertonic solutions (Markiewicz et al., 2005; Scott and O'Hare, 2001); lamin LIII from *X. laevis* oocyte nucleus is a B-type lamin.

We would like to point that the MEF experiments are not the focus of this manuscript. Moreover, we believe that the fact that chromatin is attached to lamin means that one cannot measure explicitly lamin filaments. The MEF part was added at the request of reviewer #1.

The argument that this does not happen for the *C. elegans* lamins is not enough to explain the case for MEFs.

I recommend that the author stay true to the data presented in this manuscript and should not stress that the procedure does not affect the lamina in MEFs. Therefore, the sentence in the discussion: 'Moreover, the structure of lamin filaments studied in low salt buffer was found to be identical to the filaments visualized in situ', is incorrect considering the presented data.

Although lamin filaments are observed both in MEFs and in *C. elegans*, we have removed this statement from the **Discussion** as this may require more high-resolution data. We agree with the reviewer.

We mention in the revised manuscript that the main point is that the lamin meshwork analysis provides a physical model that can be used to understand the mechanical properties of lamin filaments in the meshwork. We have mentioned this in the **Discussion** of the revised manuscript (**page 13**).

We have toned down the extrapolation from *in vitro* to *in vivo*. We discuss that our system is not entirely *in situ* as we are removing the nucleus from the cell and this changes the nuclear environment and may alter the mechanical properties (**page 13**). We have now minimized the use of *in situ* and also changed *in situ* to *in vitro* depending on the context. We use the term ‘*in situ* assembled lamin’ at a few places which is correct and should not be a matter of any further concern. Also, we mention clearly in the caption to **Figure 7** that the model is based on *in vitro* AFM pushing experiments and *in vivo* validity will require further investigation.

The authors state now that they have done measurements on both orthogonally assembled and randomly organized lamina meshwork. Yet, it is still interesting that they do not observe differences in the mechanical behaviour of the network. Orthogonally organized meshwork indicates a level of self-assembly and intermolecular interactions beyond that of a tetramer that I expect to contribute to the mechanical behaviour of the meshwork. I find this point weakly addressed by the authors. Currently, they only state that measurements were done on both architectures in the legend of figure 1. In the main text, the authors state ‘lamin filaments were arranged in a meshwork exhibiting a rectangular pattern or a less organized architecture interspersed and interacting with the NPCs’ (page 5) and no description of this is provided in the network topology analysis. I understand the authors do not see a difference between the oocyte and the MEF architectures, but I would strongly recommend that the authors make it clear that both architectures also resulted in similar topologies (if that is the case), although I would have liked to see an analysis of the different architectures done independently and then a quantitative comparison between them.

We appreciate the reviewer’s concern: this was indeed the reason why we did the network analysis as we explain further. The orthogonal and random lamin meshworks is an oversimplified portrayal of the meshwork, at least from AFM images (observed here) and the SEM images (Aebi et al., 1986). In the Aebi *et al.* 1986 paper, the term used is ‘near-orthogonal’. We have used the same term now in the revised manuscript.

The AFM and SEM images are 2-dimensional giving the impression that areas of meshwork are exclusively near-orthogonal or random as only the upper layer is imaged. Our cryo-ET analysis indicates that areas of near-orthogonal (apparently organized) and less organized regions are mixed. The AFM measurements were performed on such a meshwork and not only on what is seen in the 2D image. We have mentioned this now in the captions to **Figure 1** and **Supplementary Figure 2**. In tomograms from cryo-ET it is difficult to discern if the meshwork is orthogonal or random as the tomograms provide a 3-dimensional view, and the areas are overlapping creating a meshwork. **Hence the network analysis to bring out the hidden rules in the meshwork. The point of the network analysis was to determine the topology of the sub-volume of the lamin meshwork instead of biasing the result based on two meshwork types only.** We have now mentioned this in the revised manuscript (**page 10**).

We would like to emphasize that the overall mechanical behavior from *in vitro* AFM measurements resembles well the MD simulations that are done on a physical model derived

from *in vitro* structural data. In both cases, it is a meshwork or overlapping filaments as explained in the paragraph above.

In the section ‘Mechanics of mammalian nuclei’, the authors start by stating that the concentration of lamins would alter the mechanical properties of the NE. However, it is far from being clear from their experiments of the knockouts that the overall concentrations are per se altered. In addition, the knockouts do not necessarily make the nuclei softer as would be hypothesised based on the contribution of concentrations only. I believe that authors should rephrase the paragraph to focus on the potential contribution of each of the lamin types to a specific network topography (as they now show A-type lamin knockout MEF cells present orthogonal arrangement, while the B-type lamin knockout show random arrangement (Supplementary Fig. 1)).

We thank the reviewer for suggesting this and have revised the paragraph. (1) We have removed the statement quoted by the reviewer. (2) We mention that nuclear mechanics may be influenced by the **relative** concentrations of major lamins. (3) We discuss meshwork arrangement as suggested by the reviewer.

Statistical non-significance and the confidence values should still be provided despite of the large standard deviations. That is the whole point of using statistical tests to provide a confidence level in the claimed significance or insignificance. I would urge the authors to include a similar analysis in their Supplementary Fig. 20 to support their claims.

We thank the reviewer for the comment and have modified the analysis accordingly. Kruskal-Wallis ANOVA test (Origin Pro) was conducted to determine the differences between the wild-type, Lmn B dko and Lmn A ko nuclei under different compression heights. At the 0.05 level, significant differences were found among the 3 independent nuclei populations compressed to 5 μm ($p = 2.9 \times 10^{-4}$) (**b**) and 3 μm ($p = 7.5 \times 10^{-4}$) (**c**) but not among those compressed to 10 μm ($p = 0.17$) (**a**). The nuclei heights (**d**) were also significantly different ($p = 0.006$). We have reported this in the caption to **Supplementary Fig. 20**.

Minor comments:

Sup. Note 1: first paragraph, equation is incomplete: [$\theta = \tan^{-1} (\text{dlow} / (0.5 \text{ l}))$].

We do not get why the reviewer mentions it is an incomplete equation. We apologize for this misunderstanding but would be good to point it.

In Supplementary Fig. 11. Authors should clearly indicate what the color bar represents. presumably normalized densities of the populations?

The reviewer is correct that it is normalized densities of the populations. We have now mentioned that the color bar represents normalized densities of the populations in the figure legend of **Supplementary Fig. 11**.

References

- 1) Aebi, U., Cohn, J., Buhle, L., and Gerace, L. (1986). The nuclear lamina is a meshwork of intermediate-type filaments. *Nature* 323, 560-564.
- 2) Liu, H., Wen, J., Xiao, Y., Liu, J., Hopyan, S., Radisic, M., Simmons, C.A., and Sun, Y. (2014). *In situ* mechanical characterization of the cell nucleus by atomic force microscopy. *ACS Nano* 8, 3821-3828.
- 3) Markiewicz, E., Ledran, M., and Hutchison, C.J. (2005). Remodelling of the nuclear lamina and nucleoskeleton is required for skeletal muscle differentiation in vitro. *J Cell Sci* 118, 409-420.
- 4) Moir, R.D., Yoon, M., Khuon, S., and Goldman, R.D. (2000). Nuclear lamins A and B1: Different pathways of assembly during nuclear envelope formation in living cells. *J Cell Biol* 151, 1155-1168.
- 5) Scott, E.S., and O'Hare, P. (2001). Fate of the inner nuclear membrane protein lamin B receptor and nuclear lamins in herpes simplex virus type 1 infection. *J Virol* 75, 8818-8830.
- 6) Stanley, G.J., Fassati, A., and Hoogenboom, B.W. (2018). Atomic force microscopy reveals structural variability amongst nuclear pore complexes. In *Life Sci Alliance*.
- 7) Stephens, A.D., Banigan, E.J., Adam, S.A., Goldman, R.D., and Marko, J.F. (2017). Chromatin and lamin A determine two different mechanical response regimes of the cell nucleus. *Mol Biol Cell* 28, 1984-1996.

REVIEWER COMMENTS

Reviewer #1 (Remarks to the Author):

The authors have adequately addressed all my previous comments.

Reviewer #5 (Remarks to the Author):

Reviewer report on “Nonlinear mechanics of lamin filaments and the meshwork topology build an emergent nuclear lamina” by K. Tanuj Sapra et al. (2020)

In this manuscript the authors combine different mechanical and structural strategies to study the nonlinear behavior of individual lamin LIII filaments in a more physiological relevant native meshwork organization. The authors combine in vitro and in silico approaches (atomic force microscopy, cryo-electron tomography, and multiscale molecular dynamics simulations) to study the nonlinear meshwork mechanics of lamin LIII filaments from *Xenopus laevis* oocyte isolated nucleus. The authors observed by pushing and bending of “individual” lamin LIII filaments withing the exposed nuclear lamina using AFM quasi-static force curves a nonlinear strain-stiffening behavior. Moreover, the authors use in silico force curve lamin LIII deformation simulation and found similarities between the resulting force curves, suggesting that they are proving individual lamin filaments and that they do have nonlinear strain-stiffening behavior. Additionally, the authors observed that energy is dissipated during the mechanical deformation of lamin LIII filaments which suggest that lamins function as shock absorbers to protect the nuclear contents from external mechanical forces. Finally, by molecular dynamics simulations they showed that lamins in an interconnected meshwork are capable to withstand very large deformations, thus reporting a very important and emergent material characteristic of the lamina network. The manuscript is well written, and the experiments and data analysis seem to be performed carefully.

Broadly speaking, I see a clear contribution of this paper towards understanding the nonlinear mechanical properties behavior of individual lamin filaments and how this mechanics in an interconnected network (lamina) generates an emergent nonlinear mechanical behavior critical to protect the nucleus internal content from a wide variety or external forces. The presented research work is interesting, and I fell is of broad interest to the Nature Communications readership and I may recommend its publication after the authors address the following comments:

Comments:

Since this revised version of the manuscript has been already reviewed by other referees and it was sent to me to be viewed by a technical expert in atomic force microscopy, I will provide comments specifically of the use of AFM and its data analysis.

1) In this manuscript the JPK’s Quantitative Imaging (QI) method –fundamentally a fast force-volume method- was used mostly with spatial resolution of 128 X 128 pixels, thus for an imaged sample area of 1 μ mX1 μ m the spatial lateral resolution is around 8nm. Utilizing the authors lamin filaments

thickness results by Cryo-ET, the diameter of filaments is between 4-7nm, thus the filament thickness is similar or below the AFM imaging spatial resolution. This will make almost impossible to position the AFM probe with 10nm size tip (also larger than individual lamin filaments) on the center of the lamin LIII filament. If the tip is not positioned in the center (sphere-cylinder contact mechanics) of the lamin filament to vertically deflect the filament, the filament will experience a vertical and lateral force that will affect some of the observations presented in this manuscript. Could the authors comment about this?

2) Additionally, by not having the probe positioned in the filament cylindrical shape top surface center, the vertical and lateral forces could potentially increase probe slippage. This slippage could have significant impact in understanding the resulting "peaks". Potentially the first peak could be from the lamin filament buckling and failure, however the other peaks could presumably be probe tip slippage.

3) Finally, to judge about the applicability of the linear regression in force-indentation curves to determine the stiffness at different curve regions (for example to determine the stiffness values provided in figure 2), the authors should more explicitly describe and provide an example with all the stiffness fits and their goodness of fit. In the material and methods, they state they determined the stiffness manually using Punnias 3D (What is Punnias 3D? any reference?). A more detailed description should be included in the materials and methods section or supplementary information. I suggest the authors should add extracted force curves with superimposed fitted linear fits at the different force curves locations/regions and indicate the goodness of fit as supplementary figures.

Reviewer #7 (Remarks to the Author):

The authors have done a good job responding to the reviewers, I think.

In considering / reviewing the computational components, they seem substantial and well-considered. However, as written there is not enough information to reproduce the work. For example, I could not find mention of which program the authors used to carry out the work, which force fields, etc. Are the programs used "home made" or are they community codes?

Standard for the field is moving towards pushing authors to deposit, along with their manuscript, as Supporting Information, machine readable input and run files for computational work. In this way, additional parameters not mentioned by the authors in the written Methods description could still be tracked and found. I would encourage the authors to provide such files.

The other thing I didn't notice was any discussion of error / statistical or otherwise, in the context of the computational components.

I say this noting that it seems to me that the computational work is mostly a minor player in this work (though important for conclusions).

Re: Nonlinear mechanics of lamin filaments and the meshwork topology build an emergent nuclear lamina

We would like to thank the reviewers again for the constructive comments and time. We accepted all the comments and requests made by the reviewers and have modified the manuscript accordingly.

Point by point answers to Reviewers' comments:

Reviewer #1 (Remarks to the Author):

The authors have adequately addressed all my previous comments

We thank the reviewer for taking the time to review our paper and raising important points to make it better.

Reviewer #5 (Remarks to the Author):

Reviewer report on “Nonlinear mechanics of lamin filaments and the meshwork topology build an emergent nuclear lamina” by K. Tanuj Sapra et al. (2020)

In this manuscript the authors combine different mechanical and structural strategies to study the nonlinear behavior of individual lamin LIII filaments in a more physiological relevant native meshwork organization. The authors combine in vitro and in silico approaches (atomic force microscopy, cryo-electron tomography, and multiscale molecular dynamics simulations) to study the nonlinear meshwork mechanics of lamin LIII filaments from *Xenopus laevis* oocyte isolated nucleus. The authors observed by pushing and bending of “individual” lamin LIII filaments within the exposed nuclear lamina using AFM quasi-static force curves a nonlinear strain-stiffening behavior. Moreover, the authors use in silico force curve lamin LIII deformation simulation and found similarities between the resulting force curves, suggesting that they are proving individual lamin filaments and that they do have nonlinear strain-stiffening behavior. Additionally, the authors observed that energy is dissipated during the mechanical deformation of lamin LIII filaments which suggest that lamins function as shock absorbers to protect the nuclear contents from external mechanical forces. Finally, by molecular dynamics simulations they showed that lamins in an interconnected meshwork are capable to withstand very large deformations, thus reporting a very important and emergent material characteristic of the lamina network. The manuscript is well written, and the experiments and data analysis seem to be performed carefully.

Broadly speaking, I see a clear contribution of this paper towards understanding the nonlinear mechanical properties behavior of individual lamin filaments and how this mechanics in an interconnected network (lamina) generates an emergent nonlinear mechanical behavior critical to protect the nucleus internal content from a wide variety or external forces. The presented research work is interesting, and I fell is of broad interest to the Nature Communications readership and I may recommend its publication after the authors address the following comments:

We thank the reviewer for carefully reading the manuscript and the positive outlook of our work.

Comments:

Since this revised version of the manuscript has been already reviewed by other referees and it was sent to me to be viewed by a technical expert in atomic force microscopy, I will provide comments specifically of the use of AFM and its data analysis.

1) In this manuscript the JPK’s Quantitative Imaging (QI) method –fundamentally a fast force-volume method- was used mostly with spatial resolution of 128 X 128 pixels, thus for an imaged sample area of 1 μ m x 1 μ m the spatial lateral resolution is around 8nm. Utilizing the authors lamin filaments thickness results by Cryo-ET, the diameter of filaments is between 4-7nm, thus the filament thickness is similar or below the AFM imaging spatial resolution. This will make almost impossible to position the AFM probe with 10nm size tip

(also larger than individual lamin filaments) on the center of the lamin LIII filament. If the tip is not positioned in the center (sphere-cylinder contact mechanics) of the lamin filament to vertically deflect the filament, the filament will experience a vertical and lateral force that will affect some of the observations presented in this manuscript. Could the authors comment about this?

We agree with the reviewer that we cannot position the cantilever tip in the center of the filament. By ‘positioned at random points’ (as mentioned on **pg. 5**) we did not mean that we *accurately* brought the cantilever tip in the center of the lamin filament but on the filament because the measurements were performed with the closed-loop. We apologize for not making this clear. We have removed the words ‘the cantilever was positioned’, to circumvent any ambiguity and have now clarified the procedure. We also further elaborate in the **Methods (pg. 17)** the exact manner the filaments were pushed as the reviewer has suggested.

We would like to mention that the nominal tip diameter is at least twice as large (20 nm) as the diameter of the lamin filaments (~8 nm). This will enable the tip to push the entire diameter of the filament and not only on the center. However, as the reviewer pointed out the cantilever may still slip. We have mentioned this in the revised text (**pg. 5**).

It should also be noted that laterally pushing intermediate filaments *in vitro* required forces of 3 – 5 nN to break them (Kreplak, Bär et al. 2005) similar to the forces measured in our experiments. We consistently observed that the filaments could be pushed to forces of up to a 3 – 5 nN. We had mentioned this on **pg. 7** but have further elaborated on this in the revised text. However, we cannot distinguish between the vertical and lateral forces in our measurements. As suggested by the reviewer we have discussed all these points in the revised text (**pg. 5, 7**).

2) Additionally, by not having the probe positioned in the filament cylindrical shape top surface center, the vertical and lateral forces could potentially increase probe slippage. This slippage could have significant impact in understanding the resulting “peaks”. Potentially the first peak could be from the lamin filament buckling and failure, however the other peaks could presumably be probe tip slippage.

We agree with the reviewer that tip slippage from the filament may occur. We now mention this in the revised manuscript (**pg. 8**). We also always mention ‘apparent failure’ of the filament as the peak could be because of slippage. However, we also suggest that the filament can tolerate at least the measured force. In fact, if slippage would occur frequently our measured force would be an under-estimate of the strength of the filament. We have emphasized this in the revised text.

The reviewer is correct that the other peaks could be because of the tip slippage and we have mentioned this in the revised text (**pg. 9**). We would like to point out that the subsequent peaks were all observed at similar forces, in the nano Newton range. The existence of subsequent peaks with decreasing frequency (**Supplementary Fig. 14, Supplementary Table 3**) suggests that the 1st peak and also the other peaks are because of lamin failure as it is unlikely that the tip slips multiple times from the same lamin filament.

3) Finally, to judge about the applicability of the linear regression in force-indentation curves to determine the stiffness at different curve regions (for example to determine the stiffness values provided in figure 2), the authors should more explicitly describe and provide an example with all the stiffness fits and their goodness of fit. In the material and methods, they state they determined the stiffness manually using Punias 3D (What is Punias 3D? any reference?). A more detailed description should be included in the materials and methods section or supplementary information. I suggest the authors should add extracted force curves with superimposed fitted linear fits at the different force curves locations/regions and indicate the goodness of fit as supplementary figures.

As suggested by the reviewer, a supplementary figure of extracted force curves showing the stiffness fits with goodness of fits (R values) (**Supplementary Fig. 9**) as well as the software details have been added to the Methods (**pg. 17**).

Reviewer #7 (Remarks to the Author):

1) The authors have done a good job responding to the reviewers, I think.

We thank the reviewer for the time and the positive outlook.

2) In considering / reviewing the computational components, they seem substantial and well-considered. However, as written there is not enough information to reproduce the work. For example, I could not find mention of which program the authors used to carry out the work, which force fields, etc. Are the programs used "home made" or are they community codes?

Standard for the field is moving towards pushing authors to deposit, along with their manuscript, as Supporting Information, machine readable input and run files for computational work. In this way, additional parameters not mentioned by the authors in the written Methods description could still be tracked and found. I would encourage the authors to provide such files.

The programs are all commercially available and have also been used by us. We have now mentioned this in the Methods (**pg. 20**) and cited the relevant references. The authors will be delighted to provide the original data and files upon request. This is now stated in the **Code availability** section.

The other thing I didn't notice was any discussion of error / statistical or otherwise, in the context of the computational components.

I say this noting that it seems to me that the computational work is mostly a minor player in this work (though important for conclusions).

Sentences on the statistics and potential errors of the computational part (namely the MD simulations section) have been added to the revised manuscript (**pg. 12**) and in the caption to Fig. 6 (**pg. 36**).

References

Kreplak, L., et al. (2005). "Exploring the mechanical behavior of single intermediate filaments." J Mol Biol **354**(3): 569-577.

REVIEWERS' COMMENTS

Reviewer #5 (Remarks to the Author):

The authors have adequately addressed the comments I raised. I believe the manuscript has been improved. I'm satisfied with the revised manuscript and recommend proceeding with publication.

Reviewer #7 (Remarks to the Author):

The additional information provided for the molecular dynamics simulation really is still insufficient. The authors still don't even mention the force field used. A brief one paragraph summary of what was carried out here would be appropriate. If the work was wholly published in the earlier cited studies, then that is another issue and thus the computational work should be removed.

Otherwise seems fine to move ahead!

To the editors:

Re: Nonlinear mechanics of lamin filaments and the meshwork topology build a hierarchical nuclear lamina.

GENERAL COMMENTS:

We would like to thank you and all the reviewers that participated in the rounds of revisions which made the manuscript better for publishing in *Nature communication*.

REVIEWERS' COMMENTS

Reviewer #5:

The authors have adequately addressed the comments I raised. I believe the manuscript has been improved. I'm satisfied with the revised manuscript and recommend proceeding with publication.

We thank the reviewer for the support and appreciation that the work can be accepted for publication in *Nature Communications*.

Reviewer #7:

The additional information provided for the molecular dynamics simulation really is still insufficient. The authors still don't even mention the force field used. A brief one paragraph summary of what was carried out here would be appropriate. If the work was wholly published in the earlier cited studies, then that is another issue and thus the computational work should be removed.

Otherwise seems fine to move ahead!

We thank the reviewer for the support and appreciation that the work can be accepted for publication in *Nature Communications*.

A paragraph is inserted in Methods summarizing what was carried out in molecular dynamics (highlighted yellow).